# Agent-based model demonstrates the impact of nonlinear, complex interactions between cytokines on muscle regeneration

Megan Haase[1], Tien Comlekoglu[1], Alexa Petrucciani[2], Shayn M Peirce[1], Silvia S Blemker[1]*

[1]University of Virginia, Charlottesville, United States; [2]Purdue University, West Lafayette, United States

*For correspondence:
ssblemker@virginia.edu

**Abstract** Muscle regeneration is a complex process due to dynamic and multiscale biochemical and cellular interactions, making it difficult to identify microenvironmental conditions that are beneficial to muscle recovery from injury using experimental approaches alone. To understand the degree to which individual cellular behaviors impact endogenous mechanisms of muscle recovery, we developed an agent-based model (ABM) using the Cellular-Potts framework to simulate the dynamic microenvironment of a cross-section of murine skeletal muscle tissue. We referenced more than 100 published studies to define over 100 parameters and rules that dictate the behavior of muscle fibers, satellite stem cells (SSCs), fibroblasts, neutrophils, macrophages, microvessels, and lymphatic vessels, as well as their interactions with each other and the microenvironment. We utilized parameter density estimation to calibrate the model to temporal biological datasets describing cross-sectional area (CSA) recovery, SSC, and fibroblast cell counts at multiple timepoints following injury. The calibrated model was validated by comparison of other model outputs (macrophage, neutrophil, and capillaries counts) to experimental observations. Predictions for eight model perturbations that varied cell or cytokine input conditions were compared to published experimental studies to validate model predictive capabilities. We used Latin hypercube sampling and partial rank correlation coefficient to identify in silico perturbations of cytokine diffusion coefficients and decay rates to enhance CSA recovery. This analysis suggests that combined alterations of specific cytokine decay and diffusion parameters result in greater fibroblast and SSC proliferation compared to individual perturbations with a 13% increase in CSA recovery compared to unaltered regeneration at 28 days. These results enable guided development of therapeutic strategies that similarly alter muscle physiology (i.e. converting extracellular matrix [ECM]-bound cytokines into freely diffusible forms as studied in cancer therapeutics or delivery of exogenous cytokines) during regeneration to enhance muscle recovery after injury.

## eLife assessment

This is so-far the most comprehensive, spatially resolved in 2D, dynamical, multicellular model of murine muscle regeneration after injury. The work is an attempt to combine many contributors to muscle regeneration into one coherent calibrated framework. The presented analysis is **solid** and the model has the potential to be a very **valuable** tool in the areas of tissue morphogenesis, regenerative therapies, quantitative modeling and simulation.

## Introduction

Skeletal muscle injuries account for more than 30% of all injuries and are one of the most common complaints in orthopedics (*Quintero et al., 2009*; *Barroso and Thiele, 2011*; *Valle, 2011*). The standard treatment for muscle injuries is limited mostly to rest, ice, compression, elevation, anti-inflammatory drugs, and immobilization (*Quintero et al., 2009*). These treatments lack a firm scientific basis and have varied outcomes, some resulting in incomplete functional recovery, formation of scar tissue, and high injury recurrence rates (*Järvinen et al., 2007*; *Huard et al., 2022*). Our fundamental understanding of the individual cellular and subcellular behaviors of muscle cells has advanced and made it clear that interactions between cells and their microenvironment is critical for healthy regeneration. These interactions are dynamic, involve feedback mechanisms, and lead to complex emergent phenomena; therefore, there are numerous possible interventions that could enhance muscle regeneration.

Muscle regeneration requires an abundance of cells and cytokines to interact in a highly coordinated mechanism involving five interrelated cascading phases including degeneration, inflammation, regeneration, remodeling, and functional recovery (*Forcina et al., 2020*). Following an acute muscle injury, there is a time-dependent recruitment of neutrophils, monocytes, and macrophages to remove necrotic tissue and release factors that regulate fibroblast behavior and SSC activation, proliferation, and division (*Howard et al., 2020*). Following initial inflammatory response, fibroblasts and SSCs activate and proliferate with the macrophages shifting from their pro- to anti-inflammatory phenotype. In healthy muscle, this process would be followed by remodeling of the muscle where the fibroblasts apoptose and SSCs differentiate and fuse to repair the myofibers (*Westman et al., 2021*). Each cell involved in this process secretes cytokines that help regulate cell recruitment and chemotaxes to modulate the dynamics of the recovery. It has also been shown that the molecular events implicated in angiogenesis occur at early stages of muscle regeneration to restore microvascular networks that are crucial for successful muscle recovery (*Wagatsuma, 2007*).

There are numerous cytokines involved in muscle regeneration, many of which have been individually studied to examine their influence on muscle regeneration (*Chen et al., 2015*). These cytokines play key roles in dictating cell behaviors and are major drivers of the regeneration cascade (*Husmann et al., 1996*). The dynamics of these cytokines control many aspects of the microenvironment and altering their properties to optimize treatments has been proposed in a variety of settings (*Itoh, 2022*). Testing alterations in cytokine dynamics experimentally has proven to be complex and expensive due to difficulties in cytokine identification and quantification as well as confounding factors due to pleiotropic activities of cytokines and interactions with soluble receptors (*Ciano-Petersen et al., 2022*). These challenges make it difficult to holistically test different diffusion and decay properties for numerous cytokines (*Ferrara, 2010*). However, if we could better understand the synergistic effects of alteration in cytokines, we could design a more effective therapy for treating muscle injury.

There are over a million possible combinations of cytokine alterations, making it unrealistic to study all combinations with experiments alone. For this reason, an in silico approach is needed to fully explore the possible treatment landscape and make predictions on potential targets to enhance muscle recovery. Over the last several years, agent-based models (ABMs) of muscle regeneration have been developed to study muscle regeneration in a variety of applications (*Westman et al., 2021*; *Virgilio et al., 2018*; *Martin et al., 2016*; *Khuu et al., 2021*; *Khuu et al., 2023*; *Virgilio et al., 2021*). These models were foundational for exploring the role of SSCs in a variety of muscle milieus (*Westman et al., 2021*; *Virgilio et al., 2018*; *Khuu et al., 2021*) and for demonstrating how ABMs can be used to simulate therapeutic interventions (*Martin et al., 2016*). However, previous models employed simplistic, non-spatial representations of cytokine behaviors and properties, which limited their ability to recapitulate cytokine alterations such as injection of transforming growth factor beta (TGF-β) (*Virgilio et al., 2021*). Furthermore, these prior models did not include microvessel adaptations and dynamic extracellular matrix (ECM) properties which are crucial for understanding the altered microenvironmental state following muscle injury. These critical limitations must be addressed in order for ABMs of muscle regeneration to provide meaningful insights into treatments for muscle injury.

The goals of this work were to: (1) develop an ABM of muscle regeneration that includes cellular and cytokine spatial dynamics as well as the microvascular environment, (2) calibrate the model to capture cell behaviors from published experimental studies, (3) validate model outcomes by comparison with

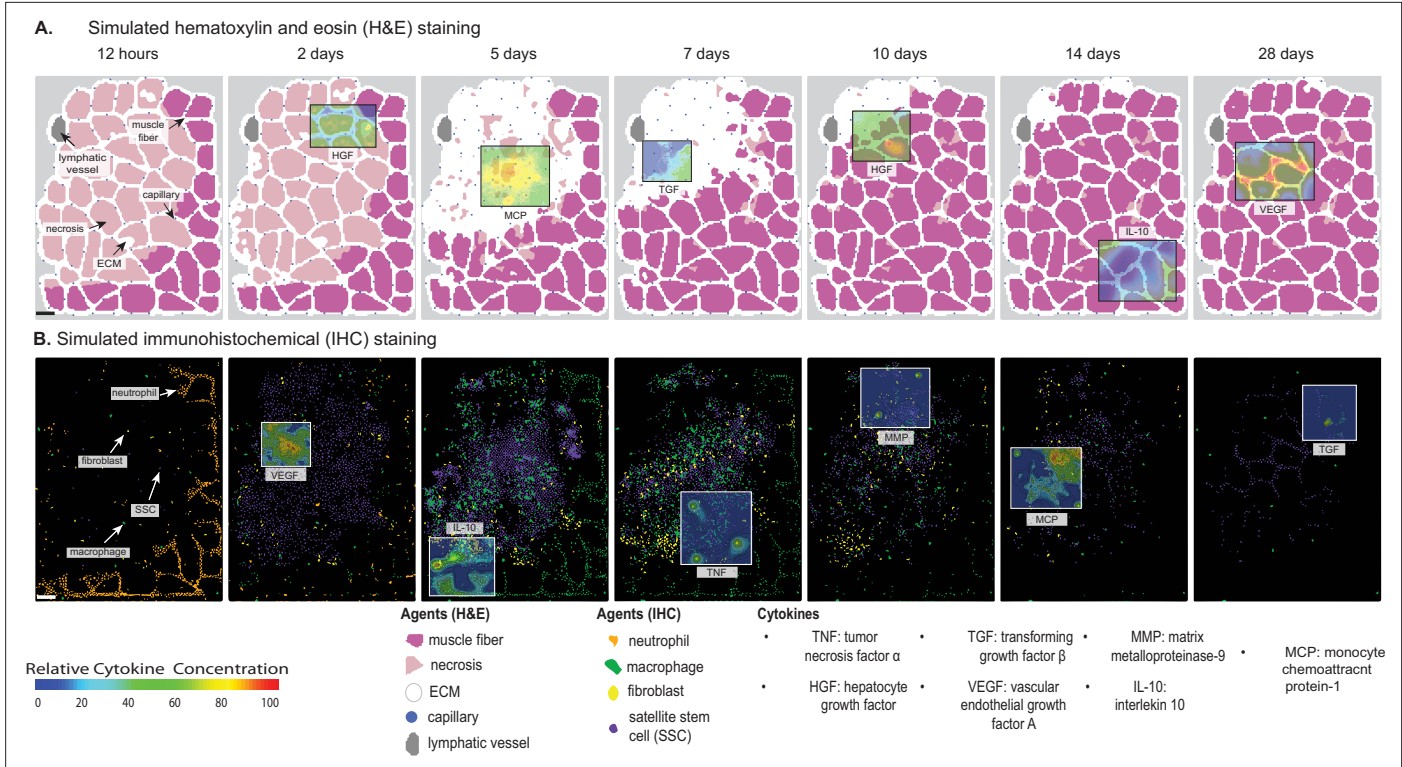

**Figure 1.** Overview of agent-based model (ABM) simulation of muscle regeneration following an acute injury. (**A**) Simulated cross-sections of a muscle fascicle that was initially defined by spatial geometry from a histology image. Muscle injury was simulated by replacing a section of the healthy fibers with necrotic elements. In response to the injury, a variety of factors are secreted in the microenvironment which impacts the behavior of the cells. The colors correspond with those typically seen in H&E staining. (**B**) ABM screen captures show the spatial locations of the cells throughout the 28-day simulation. The agent colors were matched to those typically seen in IHC-stained muscle sections. Scale bar: 50 μm.

The online version of this article includes the following figure supplement(s) for figure 1:

**Figure supplement 1.** Overview of agent-based model (ABM) simulation with different initial histology configuration.

multiple published experimental studies, (4) conduct in silico experiments to predict how altering cytokine dynamics impacts muscle regeneration. For model calibration, we implemented an iterative and robust parameter density estimation protocol to refine the parameter space and calibrate to temporal biological datasets (*Joslyn et al., 2021*). Partial rank correlation coefficient (PRCC) was used to guide in silico experiments by identifying parameters and timepoints that were most critical for ideal regeneration metrics.

## Results

### ABM outputs align with calibration and validation data

Following parameter density-based calibration, the unknown parameters were narrowed into a final calibration parameter set (*Supplementary file 1*). The simulations captured SSC and fibroblast cellular behaviors, as well as CSA outcomes, that aligned with experimental studies (*Figure 1*; *Figure 2A–C*). The model data were consistent with the experimental trends, and the 95% confidence interval was within the standard deviation (SD) for all calibration data timepoints except for SSCs at day 3 (*Figure 2B*). Macrophage (total, M1, and M2), neutrophil, and capillary counts, which were not used for model calibration, were also found to be consistent with experimental trends and allowed us to independently validate model outputs (*Figure 2D–H*).

### ABM perturbations are consistent with published experiments

Overall, the model reproduced findings from multiple studies, replicating how altered conditions lead to both improved and diminished muscle regeneration (*Figure 3*). Injections of vascular

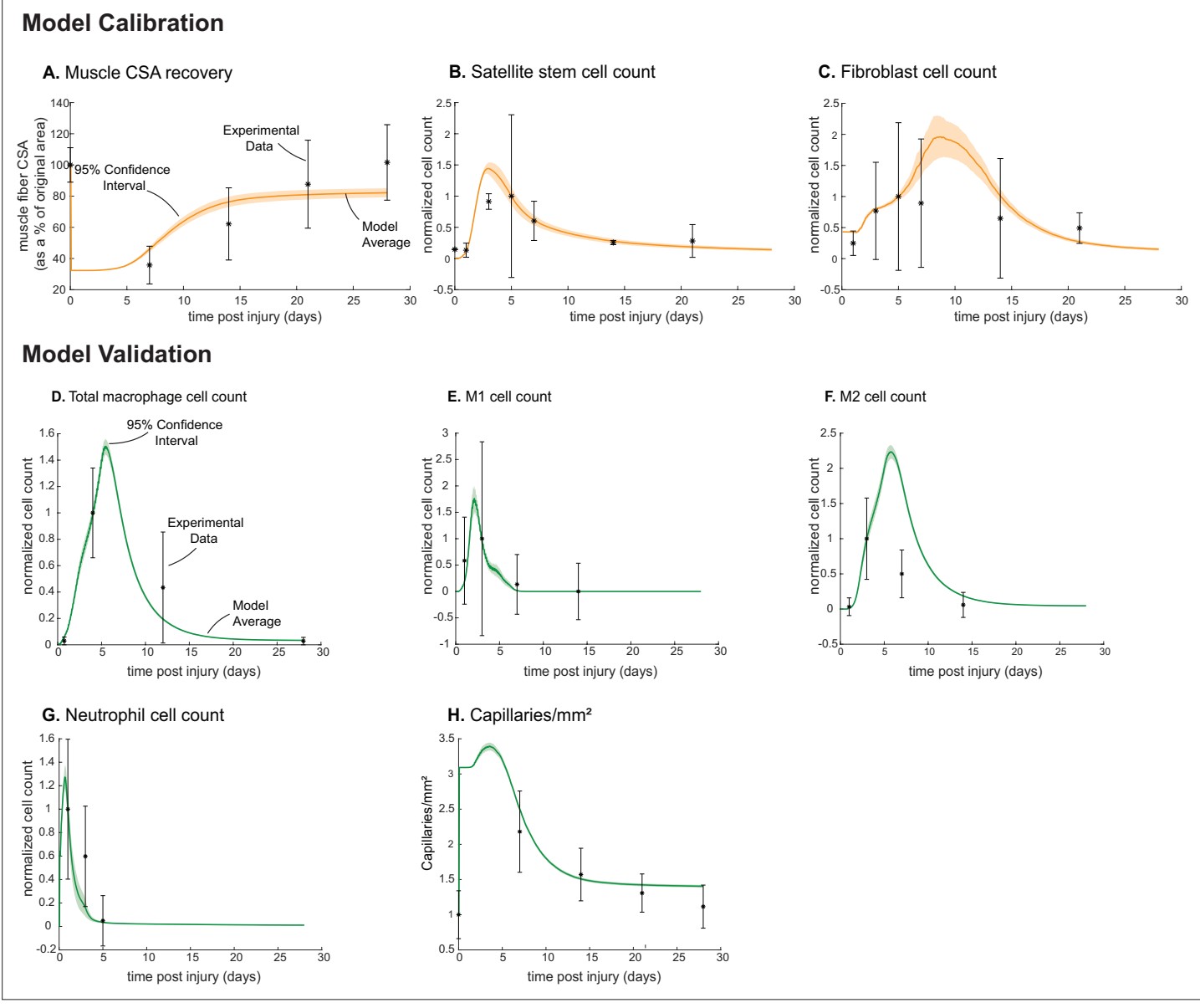

**Figure 2.** Agent-based model (ABM) calibration and validation. ABM parameters were calibrated so that model outputs for cross-sectional area (CSA) recovery, satellite stem cell (SSC), and fibroblast counts were consistent with experimental data (**A–C**). (*Murphy et al., 2011*; *Ochoa et al., 2007*). Separate outputs from those used in calibration were compared to experimental data (*Hardy et al., 2016*; *Ochoa et al., 2007*; *Wang et al., 2018*; *Nguyen et al., 2011*) to validate the ABM (**D–H**). Error bars represent experimental standard deviation, and model 95% confidence interval is indicated by the shaded region. Cell count data were normalized by number of cells on the day of the experimental peak to allow for comparison between experiments and simulations.

The online version of this article includes the following figure supplement(s) for figure 2:

**Figure supplement 1.** Overview of calibration methods.

endothelial growth factor A (VEGF-A) led to faster CSA recovery, more damaged tissue clearance, and a concentration-dependent dose response, consistent with prior studies (*Arsic et al., 2004*). Cell depletion simulations predicted decrease in all markers of regeneration, consistent with prior studies (*Arsic et al., 2004*; *Teixeira et al., 2003*; *Liu et al., 2017*). When simulating hindered angiogenesis conditions, the model aligned with experimental studies showing detriments in CSA recovery, increased neutrophil and macrophage cells, and elevated ECM collagen density, indicating progression of fibrosis within the microenvironment (*Hardy et al., 2019*). There were a few cases in which model predictions did not align with published studies. First, simulations of tumor necrosis

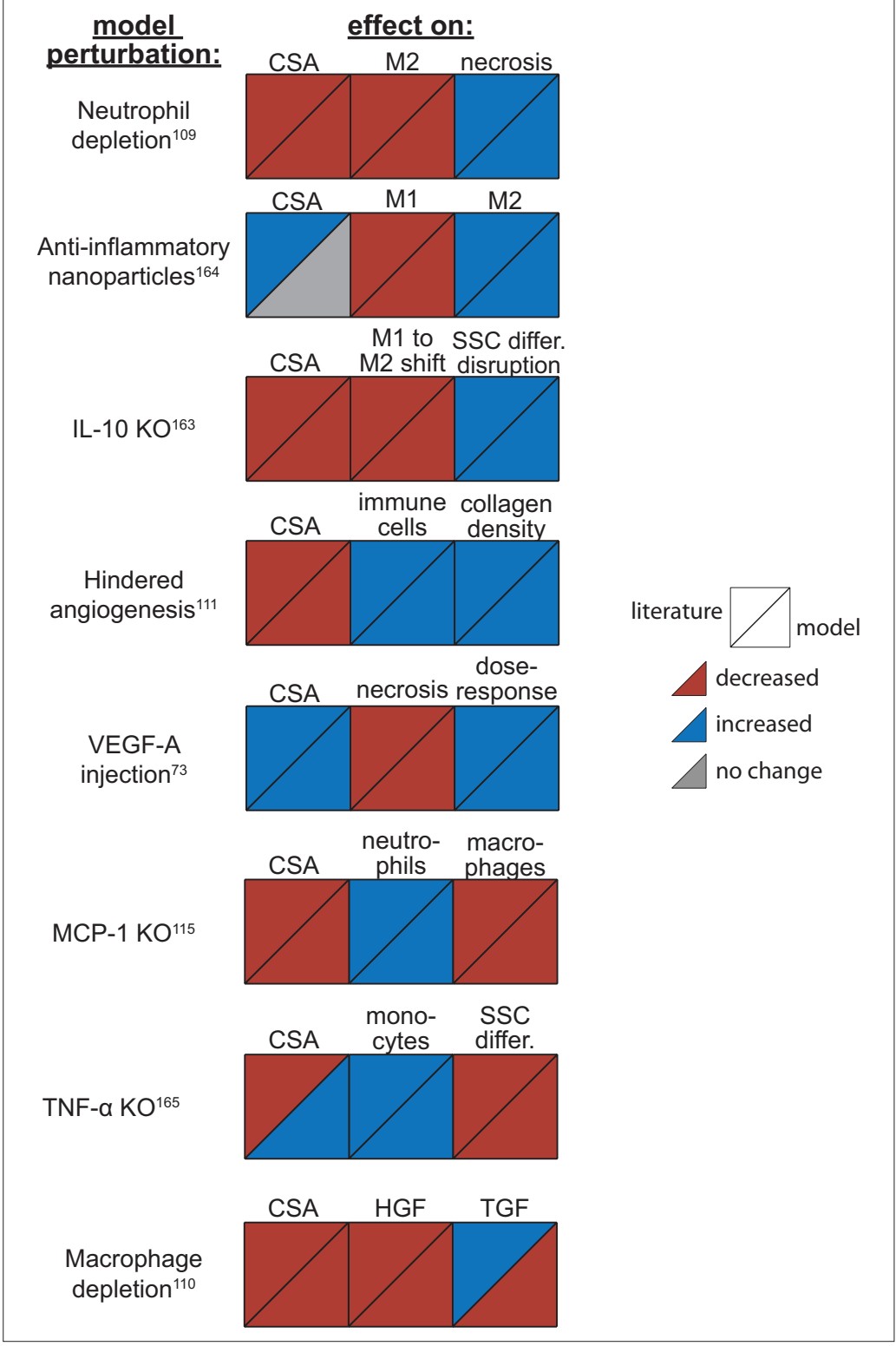

**Figure 3.** Agent-based model (ABM) perturbation outputs are compared to various literature experimental results. Each perturbation model output is compared to the available corresponding published result. The top triangles indicate the literature findings and the bottom triangles indicate the model outputs. Red triangles represent a decrease, blue represents an increase, and gray represents no significant change. Timepoints of comparison were based on which timepoints were available from published experimental data. Refer to *Table 8* for model input conditions and *Supplementary file 7* for information on experimental references.

factor alpha (TNF-α) knockout (KO) predicted increased CSA recovery, while experiments measured decreased recovery of CSA. This difference is likely due to the fact that the model did not include cross-regulation with interferons which are upregulated with TNF-α KO (*Cantaert et al., 2010*). Second, macrophage depletion simulations predicted decreased TGF-β concentrations throughout the simulation while experiments measured an initial decrease in concentration followed by increased concentrations at days 7 and 14. This difference may be due to the fact that macrophage depletion was experimentally induced with clodronate-containing liposomes which could have reduced consistency of depletion across the time course and other downstream impacts that were not represented by decreasing macrophages in the model perturbation (*Liu et al., 2017*).

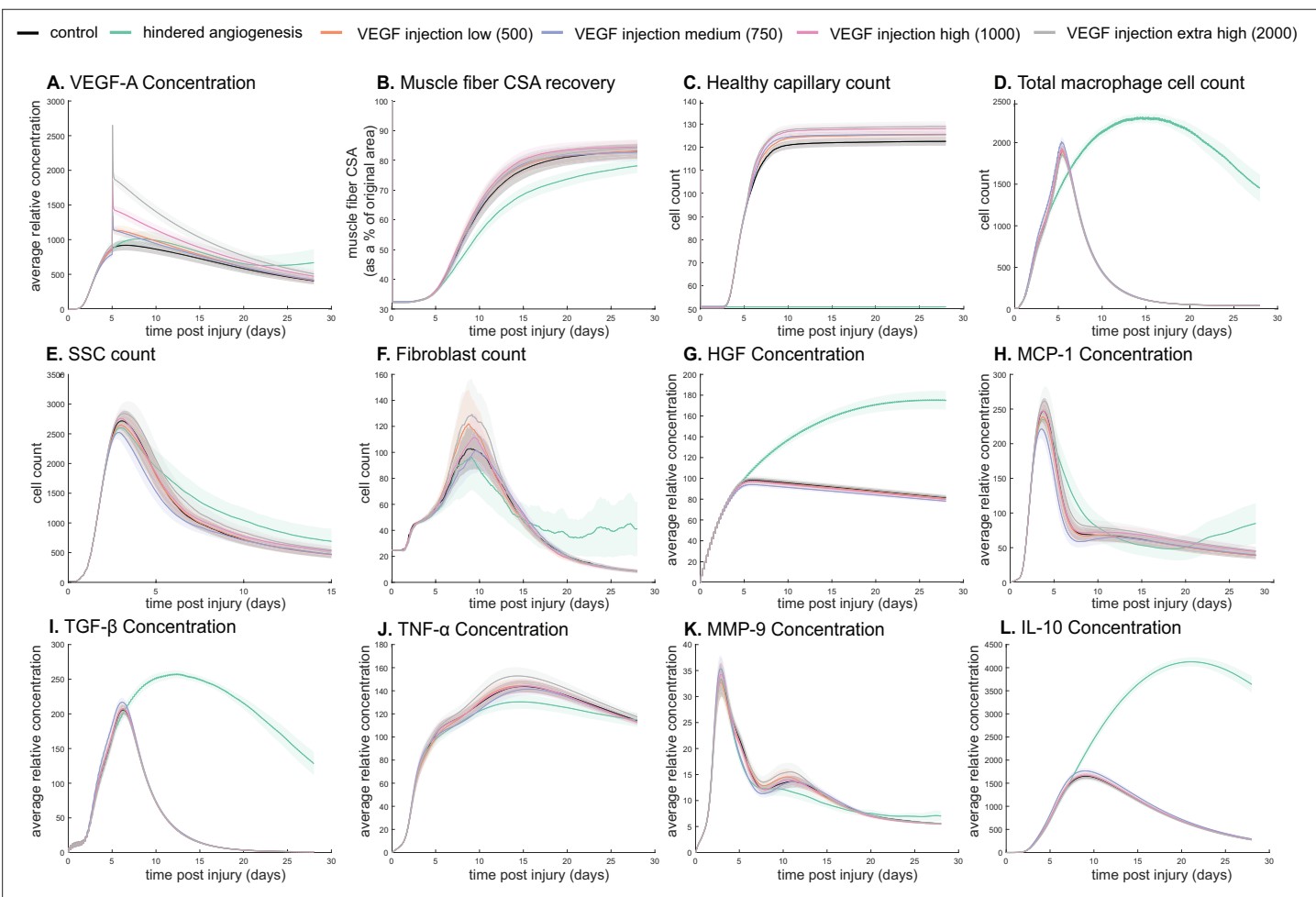

**Figure 4.** Dose-dependent response with vascular endothelial growth factor A (VEGF-A) injection compared to hindered angiogenesis. VEGF-A concentration response to varied levels of VEGF injection (**A**). Hindered angiogenesis resulted in slower and overall decreased cross-sectional area (CSA) recovery (**B**). Capillary count was dependent on VEGF-A injection level (**C**). Total macrophage count was similar between control and VEGF-A injection perturbations but macrophage count was higher in later timepoints in the hindered angiogenesis simulation (**D**). Satellite stem cell (SSC) peak varied with VEGF-A injection level and counts were prolonged in the hindered angiogenesis simulations (**E**). The fibroblast peak was lower for the hindered angiogenesis perturbation and highest with the extra high VEGF-A injection. In contrast to the other simulations, the fibroblast count was trending upward at later timepoints in the hindered angiogenesis perturbation (**F**). Hepatocyte growth factor (HGF) levels were consistent between control and VEGF-A injection perturbations but was significantly elevated in the hindered angiogenesis perturbation (**G**). Monocyte chemoattractant protein-1 (MCP-1), transforming growth factor beta (TGF-β), and interleukin 10 (IL-10) concentrations were elevated at later stages of regeneration with hindered angiogenesis (**H, I, L**). Tumor necrosis factor alpha (TNF-α) was elevated with the extra high VEGF-A injection and lower with hindered angiogenesis (**J**). Matrix metalloproteinase-9 (MMP-9) concentration was lower at the simulation midpoint but elevated at late regeneration stages (**K**).

## Analysis of ABM perturbations leads to new insights regarding cytokine and cell dynamics

The model allowed for new insights into the dynamics of muscle regeneration by providing additional timepoints and metrics to evaluate the response to exogenous delivery of VEGF-A and hindered angiogenesis. VEGF-A levels remained elevated compared to control simulations following the injection at day 5 post injury (*Figure 4A*). CSA recovery had the highest increase at 28 days post injury with the high ($10^3$ relative concentration delivered) VEGF-A injection followed by the extra high ($2 \times 10^3$ relative concentration delivered) injection (*Figure 4B*). The medium (750 relative concentration delivered) and low (500 relative concentration delivered) VEGF-A injections had higher CSA recovery 15 days post injury but were not significantly different from the control at day 28. All VEGF-A injections had a higher capillary count and were proportional to the level of VEGF-A injection (*Figure 4C*). The impact of VEGF-A injection on peak SSC and fibroblast counts was dependent on dosage amount, with the extra high VEGF-A injection resulting in the largest peaks (*Figure 4E and F*). Cytokine concentration

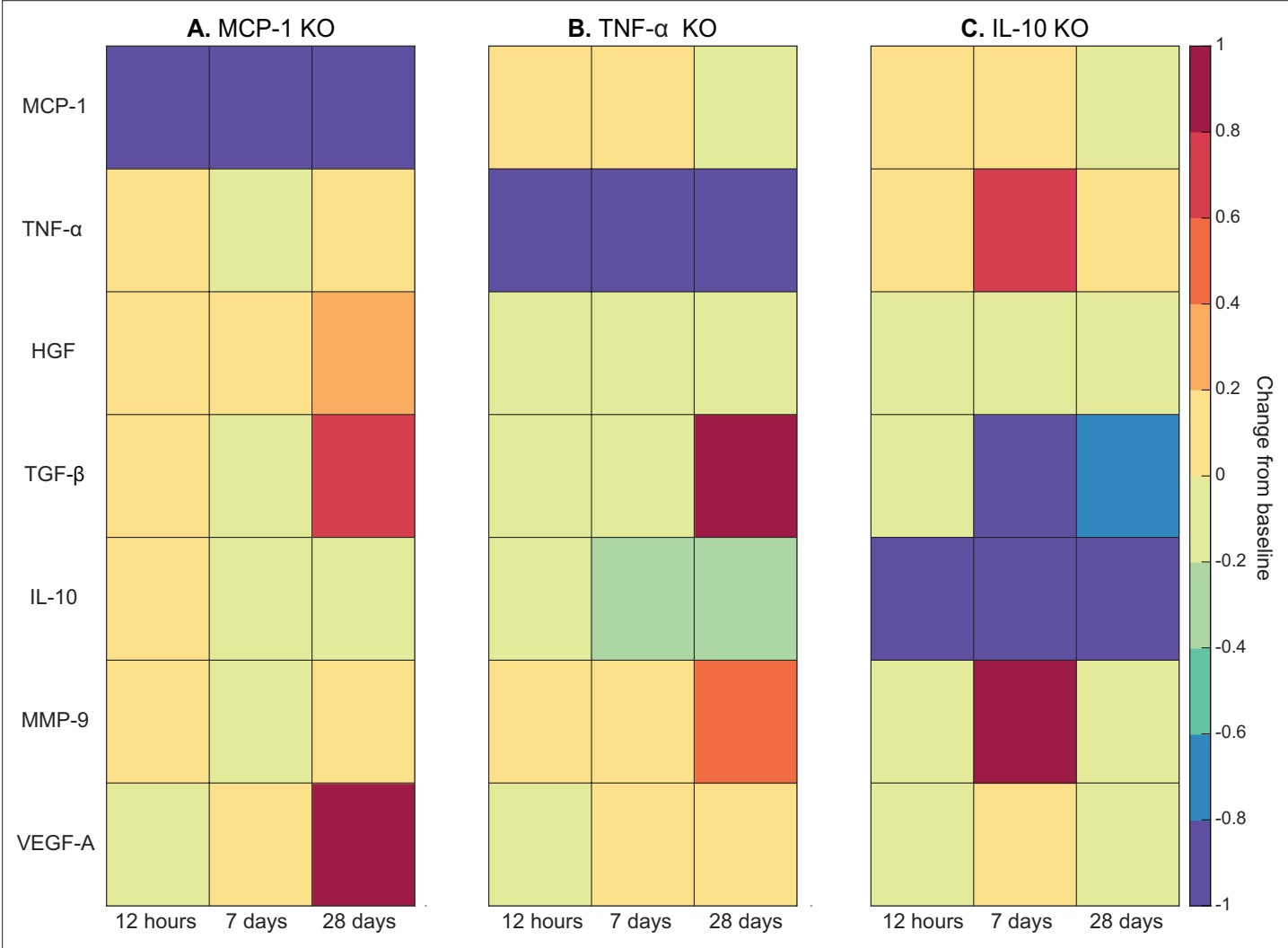

**Figure 5.** Heatmaps of changes in cytokine concentration at various timepoints throughout regeneration following individual cytokine knockout (KO) demonstrating cross-talk between cytokines. With monocyte chemoattractant protein-1 (MCP-1) KO there was an increase in all cytokines except vascular endothelial growth factor A (VEGF-A) at 12 hr post injury. Over the course of regeneration there was continued increasing elevation of hepatocyte growth factor (HGF), increases in VEGF-A, and transforming growth factor beta (TGF-β) decreased at day 7 followed by a strong increase by day 28 post injury (**A**). In the tumor necrosis factor-alpha (TNF-α) KO simulations, there was an early decrease in TGF-β that shifts to strong increases by day 28. Matrix metalloproteinase-9 (MMP-9) increased throughout the duration, HGF and interleukin 10 (IL-10) were decreased, VEGF-A lagged in the beginning but was increased during mid to late timepoints (**B**). Following IL-10 KO there were increases in TNF-α, decreases in HGF and TGF-β, and elevated MMP-9 at day 7 that decreased by day 28 (**C**).

trends were similar for all injections, but most peak levels were dosage dependent (*Figure 4G–L*). In contrast, hepatocyte growth factor (HGF) levels were elevated from days 5 to 28 with hindered angiogenesis, as were TGF-β and interleukin 10 (IL-10) (*Figure 4I and L*). Monocyte chemoattractant protein-1 (MCP-1) concentration had a lower overall peak level with elevated levels from days 21 to 28 (*Figure 4H*). Hindered angiogenesis had lower CSA recovery throughout the simulation and did not achieve unaltered regeneration levels (*Figure 4B*).

Cytokine KO perturbations revealed cross-talk and temporal interplay between cytokines (*Figure 5*). For example, with MCP-1 KO there was an overall increase in cytokine levels for all other cytokines within the microenvironment except for VEGF-A at 12 hr post injury (*Figure 5A*). By 7 days post injury TNF-α, TGF-β, IL-10, and matrix metalloproteinase-9 (MMP-9) had decreased from unaltered regeneration day 7 levels but VEGF-A and HGF were elevated. With TNF-α KO there was a decrease in TGF-β at early timepoints but a strong increase by day 28 (*Figure 5B*). Following IL-10 KO there was an increase in TNF-α that peaked at 7 days post injury (*Figure 5C*). HGF was slightly decreased throughout and TGF-β was strongly decreased by day 7. MMP-9 was decreased at 12 hr and 28 days post injury but heavily increased at day 7.

## Cytokine dynamic analysis leads to new model perturbations that predict improved regeneration

Latin hypercube sampling (LHS)-PRCC of cytokine decay and diffusion parameters elucidated temporal relationships between cytokine parameters and key regeneration metrics, such as positive correlations between CSA and TGF-β and MMP-9 decay (Table 9). Of all cytokine parameters, the model outputs were most sensitive to HGF decay, with all outputs except M1 cell count being significantly impacted. PRCC plots showed that TGF-β and MMP decay were positively correlated and HGF decay was negatively correlated with CSA recovery, with higher significance at timepoints after 12 days (*Figure 6—figure supplement 1*). Correlation plots for various cytokine concentrations and regeneration metrics showed trends in cytokine-dependent cell behaviors such as the TNF-α concentration that led to heightened fibroblast cell counts as well as the corresponding TNF-α concentration threshold that results in diminished fibroblast response (*Figure 6—figure supplement 2*). These PRCC trends guided cytokine parameter perturbations to include lower HGF and VEGF-A decay, higher TGF-β, MMP-9, and MCP-1 decay, and higher MCP-1 diffusion because each of the cytokine modifications indicated some form of enhanced regeneration outcome metrics (*Supplementary file 2*). All these perturbations except MCP-1 decay show increased CSA, increased healthy capillaries, and increased SSCs (*Figure 6*). Finally, a combination of all changes except for MCP-1 decay was simulated. The combined cytokine alteration resulted in the highest CSA recovery (*Figure 6A*), as well as increased M1 macrophage counts (*Figure 6B*), decreased M2 macrophage counts (*Figure 6C*), increased fibroblasts (*Figure 6D*) and SSCs cell counts (*Figure 6E*). Capillaries regenerated faster in the combined perturbation than under unaltered conditions (*Figure 6F*, *Figure 6—figure supplement 3*). It is likely that the combination of cytokines perturbed cell dynamics in a manner that promoted regeneration in both the early and later phases. During early regeneration, lower HGF decay, higher TGF decay, and MCP-1 diffusion contributed to increased SSCs while lowered VEGF decay increased angiogenesis. During late regeneration, lower HGF decay and higher MMP decay contributed to an increased anti-inflammatory state and SSC differentiation. The combined cytokine perturbation predicted a 13% improvement in CSA recovery compared to the unaltered regeneration amount at 28 days. The combined cytokine perturbation also had higher peaks in SSC and fibroblast counts than any of the singular cytokine perturbations, indicating the synergistic effects of altering the cytokine dynamics in combination.

## Discussion

We developed a novel ABM that recapitulates muscle regeneration and, unique from prior work, includes spatial interactions between cytokines and the microvasculature based on relevant literature (*Westman et al., 2021*; *Virgilio et al., 2018*; *Martin et al., 2016*). The creation of the model provides a more controlled environment for studying muscle regeneration, reducing error and variation commonly encountered with in vivo experiments. Model predictions aligned with experimental data under various altered inputs. Through in silico experiments, we gained new insight into how the

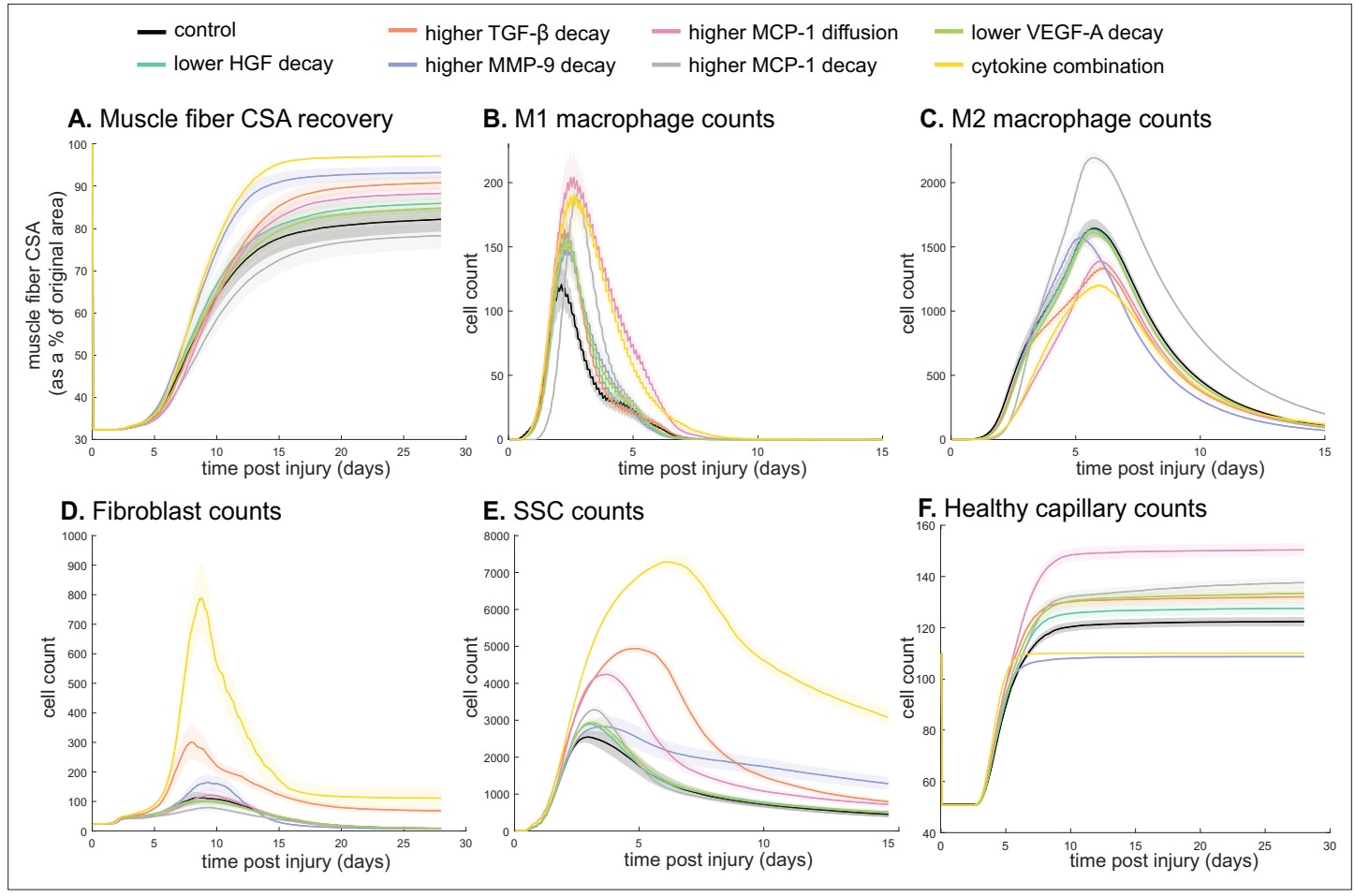

**Figure 6.** Combined alterations of various cytokine dynamics enhance muscle regeneration outcomes. All tested alterations except higher monocyte chemoattractant protein-1 (MCP-1) decay resulted in higher cross-sectional area (CSA) recovery compared to the control (**A**). M1 cell count was higher for all perturbations with the highest peaks with increased MCP-1 diffusion and the combined cytokine alteration perturbation (**B**). Higher MCP-1 decay resulted in the largest M2 peak and higher MCP-1 diffusion, higher transforming growth factor beta (TGF-β) decay, and the combined cytokine alteration had a lower M2 peak than the control (**C**). Fibroblasts had the largest increase in cell count with the higher TGF-β decay and the cytokine combination perturbations (**D**). All perturbations resulted in an increased satellite stem cell (SSC) count with the largest increase resulting from the combined cytokine alteration (**E**). All perturbations except the combined and higher matrix metalloproteinase-9 (MMP-9) decay resulted in increased capillaries as a result of additional capillary sprouts (**F**).

The online version of this article includes the following figure supplement(s) for figure 6:

**Figure supplement 1.** Partial rank correlation coefficient (PRCC) plots for various model outputs over time to illustrate how the significance of cytokine decay and diffusion parameters varies at different points throughout regeneration.

**Figure supplement 2.** Cytokine concentrations are correlated with cell counts and recovery metrics at various stages of regeneration.

**Figure supplement 3.** Non-perfused capillaries for each cytokine perturbation.

---

combination of key cytokine dynamic alterations could increase SSC cells and enhance CSA recovery. The ability for altered cytokine concentrations to change regeneration outcomes is consistent with studies that have found enhanced muscle recovery with delivery of platelet-rich plasma (PRP) which contain VEGF and TGF- β (***Kunze et al., 2019***). These model perturbations allow development of hypotheses and can provide the basis for future experiments and potential therapeutic interventions such as plasminogen activators to alter cytokines dynamics to enhance muscle recovery.

## ABM provides biological insight on nonlinear effects of cytokine levels

The ABM offers valuable insights into the muscle regeneration dynamics under various altered conditions, elucidating the complex interplay of cytokines, angiogenesis, and cell behaviors. Systematic simulations reveal critical thresholds, nonlinear effects, and synergistic cytokine combinations

impacting regeneration. Perturbations varying VEGF-A injection doses showed increased CSA recovery up to a threshold (high VEGF-A injection simulation), beyond which further improvements in CSA recovery cease. Cytokine KO simulations revealed the complex nature of the relationship between cytokines; removal of one cytokine from the system has a cascading temporal impact. Relationships between cytokines and cellular outputs exhibit nonlinear effects, as seen with the limited impact of elevated HGF on CSA recovery beyond a threshold and the non-monotonic relationship between TNF-α and fibroblast counts (*Figure 6—figure supplement 2*). Further analysis revealed that specific combinations of cytokine perturbations could enhance regeneration beyond singular cytokine interventions. For example, a combined intervention of: (1) decreasing HGF and VEGF-A decay, (2) increasing TGF-β and MMP-9 decay, and (3) increasing MCP-1 diffusion enhanced muscle regeneration. Prior studies have shown that individually, increased HGF (*Choi et al., 2019*), VEGF-A (*Arsic et al., 2004*), and MCP- (*Liu et al., 2023*) stimulate muscle regeneration whereas reduced TGF-β (*Girardi et al., 2021*) and MMP-9 (*Zimowska et al., 2012*) stimulate muscle regeneration. The model suggests that combined alterations have a stronger regenerative effect than individual cytokine changes, enhancing muscle recovery through distinct mechanisms—increasing healthy capillaries, SSC counts, and reducing inflammatory cells.

Cytokine modifications intended to enhance muscle recovery can have clinical relevance and have been studied in various settings. For example, synthetic biomaterials coated with IL-4 have been implanted as a cytokine delivery vehicle and were successful in increasing M2 cells within the muscle (*Dziki et al., 2018*). Cytokine antagonist has been successful at promoting muscle regeneration, seen in prior work with anti-IL-6 (*Fujita et al., 2014*). Studies have also shown that activation of plasmin is able to induce the release of ECM-bound VEGF, increasing angiogenesis (*Ferrara, 2010*; *Ismail et al., 2021*). Due to the complex network of cytokines, studies that deliver simple modulation of one or two cytokines typically have an insufficient response to generate appreciable improvements. This suggests that using a combination of biological and synthetic biomaterials to modulate multiple cytokines is necessary, which aligns with our findings (*Dziki et al., 2018*). Multiple cytokines have been modulated through the use of PRP which contains VEGF-A and an array of other cytokines, but PRP has had mixed success in a clinical setting (*Alsousou et al., 2013*). Our model has the capability to test and optimize various combinations of cytokines, along with exploring different temporal schedules for delivering specific treatments. For instance, it can predict whether modified combinations of cytokines prove beneficial at specific timepoints, aiding in the development of optimal treatment compositions aligned with the temporal dynamics of the regeneration cascade. These predictions provide novel concepts for future experiments and potential interventions. For example, the predictions from the model suggest that interventions that combine activation of plasmin for bound VEGF release (*Ferrara, 2010*; *Ismail et al., 2021*) with delivery of synthetic biomaterials coated with HGF (*van de Kamp et al., 2013*), TGF-β antagonist (*Akhurst, 2002*), nuclear factor-kappa B inhibitory peptide to inhibit MMP-9 (*Li et al., 2009*), and recombinant MCP-1 hydrogels (*Lin et al., 2010*) to alter diffusion rate would result in improved regeneration outcomes.

## Advancements from prior muscle regeneration models

Previous studies have employed computational models to investigate muscle regeneration across diverse contexts, such as Duchenne muscular dystrophy (DMD) and volumetric muscle loss (*Westman et al., 2021*; *Virgilio et al., 2018*). Earlier muscle regeneration ABMs from our group have been used to test the effects of priming muscle with inflammatory cells prior to injury (*Martin et al., 2016*). While these models laid the foundation for simulating muscle adaptations, they were constrained by limited diffusion capabilities and an absence of critical features related to microvessel growth and remodeling throughout the regeneration process. Similarly, other ABMs from our group have examined altered microenvironments, but their omission of spatial cytokine diffusion hindered comprehensive representation of cell behaviors pivotal to regeneration (*Westman et al., 2021*; *Virgilio et al., 2018*.) Recently, new ABMs have been published that focus on cerebral palsy and the impact of injury type on eccentric contraction-induced damage (*Khuu et al., 2021*; *Khuu et al., 2023*).

The model presented here provides advancements over prior models in three areas: (1) explicit modeling of cytokine-specific diffusion and decay that depends on the ECM environment, (2) addition of microvasculature, and (3) incorporation of a robust and rigorous calibration and validation process. The addition of microvessel growth and remodeling dynamics empowers investigations into

how interventions impact angiogenesis during regeneration, thereby influencing muscle recovery outcomes. By considering the intricate relationship between microvessels and regeneration, our model opens avenues for evaluating the effects of interventions on the broader recovery process. Second, understanding how cytokines influence cell behaviors at different times during regeneration is crucial for determining optimal treatment targets and dosing. While cytokine dynamics can be altered experimentally, doing so is expensive and time-consuming (*Itoh, 2022*; *Ferrara, 2010*) so exploring many combinations of alterations would be practically infeasible. Our model incorporates decay and diffusion dynamics of a subset of cytokines to allow testing of far more alterations in cytokines than would be reasonable to conduct experimentally. Lastly, we leveraged the CaliPro technique for parameter density estimation-based calibration and LHS-PRCC to gain biological insight by analyzing how altered microenvironmental parameters could benefit regeneration outcomes. This approach of implementing parameter identification to guide model perturbations demonstrates the capabilities of the model as a novel tool for generating new hypotheses and identifying mechanisms to target for enhanced regeneration outcomes.

Our model predictions are generally consistent with these prior models, with added biological complexity that has yielded several new important insights. For example, simulation of hindered angiogenesis predicted a decrease in SSCs leading to poor CSA recovery, similar to how lower SSC counts resulted in lower CSA recovery in perturbations in both healthy and DMD simulations (*Virgilio et al., 2018*). Our model provides additional understanding about the corresponding spatial cytokine changes that ultimately result in modulation of SSC dynamics within the microenvironment. The additional model advancements incorporated address prior muscle regeneration modeling gaps in understanding of how angiogenesis alters recovery outcomes as well as the response of complex spatial cell and cytokine dynamics.

## Limitations and future work

There are some important limitations of this study that should be discussed. First, the model does not include all cell types and cytokines that are known to influence muscle regeneration and does not account for cytokine subtype or differences between endogenous and exogenous cytokines. These cells and cytokines likely have redundant functions, given the model effectively captures muscle regeneration using the included cells and cytokines. Second, the model does not currently represent hypertrophy during regeneration, which restricts CSA recovery from surpassing 100%; however, the cell dynamics it portrays remain consistent with those observed in studies that lead to hypertrophy following injury. Third, we assume a two-dimensional (2D) cross-section based on similar ABMs that have explored the relations of 2D to 3D simulations. These studies found that the diffusion accuracy is not greatly varied and that 2D is sufficient to predict the same mechanisms seen in 3D simulations (*Marino et al., 2018*; *Sego et al., 2017*). To determine the robustness of the 2D initial cross-section, preliminary testing has shown that the initial spatial configuration can be altered and still achieve similar results (*Figure 1—figure supplement 1*), but further examination is needed to determine sensitivity to numerous configurations. Fourth, the calibration and validation dataset integrated multiple datasets from diverse sources. We acknowledge inherent limitations arising from variations in sample sizes and experimental techniques across sources. Fifth, it is also possible that the calibrated parameters are unable to capture behaviors that were not exhibited within the experimental datasets used in parameterization. While we tested ranges for each parameter and settled on a single parameter set that best fits the calibration data, there may be additional parameter sets that fit the calibration data but have varied levels of stochasticity and altered reproducibility of replicate simulations. Lastly, the current model was calibrated to male mice data despite known sex difference in skeletal muscle, regeneration mechanisms, and the timeline of recovery (*Haizlip et al., 2015*; *Knewtson et al., 2022*; *Liu et al., 2023*). Experimental measurements of female muscle regeneration are fairly limited because most muscle injury studies only use male mice or do not distinguish between sexes, making it difficult to incorporate sex differences into the model (*Enns and Tiidus, 2010*). Experiments that incorporate female mice and measure hormone levels are needed to accurately incorporate rules to distinguish between the sex-dependent dynamics of muscle regeneration.

This paper describes a significant advancement in modeling the complex process of muscle regeneration. Future efforts will extend the use of parameter density estimation to optimize the selection, doses, and timing of injections of exogenously delivered cytokines. Further refinement of analysis

methods could be pursued to disentangle specific underlying mechanisms of the dynamic feedbacks that drive the observed model outputs. Predictions from model simulations will also be used to inform future experiments by highlighting crucial timepoints to measure and predicted effect sizes for power analysis. Additionally, we aim to explore diverse muscle injury types and locations (i.e. injury relative to microvascular components) and their varying recovery responses, addressing challenges in comparing different acute injury techniques found in the literature. This study underscores the significance of cellular and cytokine spatial dynamics in muscle regeneration. Further inclusion of additional factors and hormones would provide a more holistic understanding of the system and how treatments may be altered based on microenvironmental conditions, providing a unique framework for the study of personalized muscle injury treatment.

## Materials and methods
### ABM development overview

ABMs represent the behaviors and interactions of autonomous agents, such as cells, which are governed by literature-derived rules (*Virgilio et al., 2018*; *Martin et al., 2015*; *Ferrari Gianlupi et al., 2022*). Agent-based modeling (ABM) provides an excellent platform for studying complex cellular dynamics because they reveal how the interactions between individual cellular behaviors lead to emergent behaviors in the whole system.

We implemented the ABM in CompuCell3D (version 4.3.1), a Python-based modeling software (*Swat et al., 2012*). The ABM's code is available for download (https://zenodo.org/records/10403014). To build the model, we extended upon about 40 rules developed in previous ABMs of

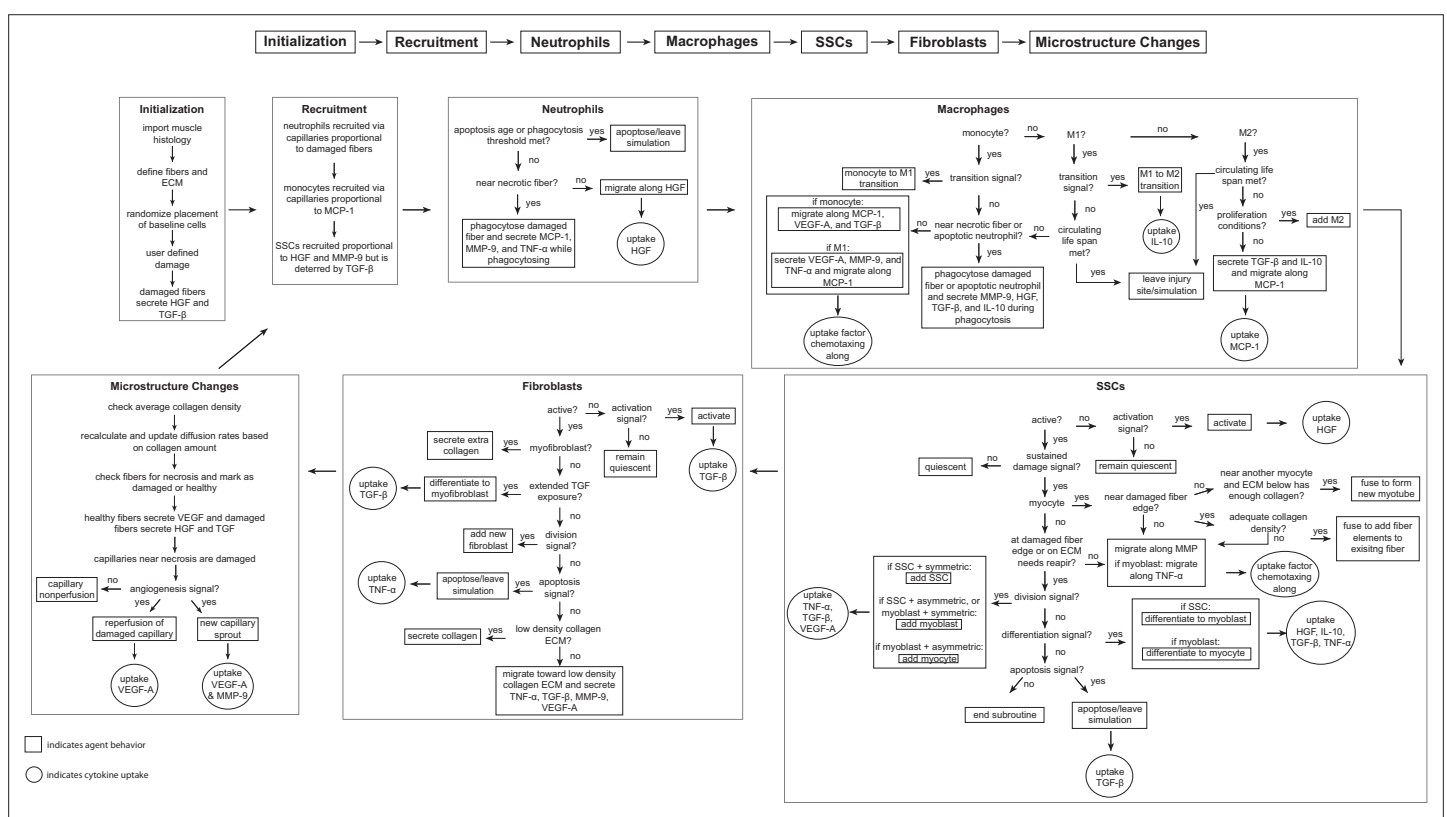

**Figure 7.** Flowchart of agent-based model (ABM) rules. The model starts with initialization of the geometry and the prescribed injury. This is followed by recruitment of cells based on relative cytokine amounts within the microenvironment. The inflammatory cells, SSCs, and fibroblasts follow their literature-defined rules and probability-based decision tree to govern their behaviors. The boxes represent the behavior that the agent completes during that timestep given the appropriate conditions and the circles represent the uptake that occurs as a result of the simulated binding with microenvironmental factors for certain cell behaviors. ABM, agent-based model; SSC, satellite stem cell; ECM, extracellular matrix; TGF-β, transforming growth factor beta; HGF, hepatocyte growth factor; TNF-α, tumor necrosis factor alpha; VEGF-A, vascular endothelial growth factor A; MMP-9, matrix metalloproteinase-9; MCP-1, monocyte chemoattractant protein-1; IL-10; interleukin 10.

muscle regeneration (*Westman et al., 2021*; *Virgilio et al., 2018*; *Martin et al., 2016*) in combination with a deep literature search referencing over 100 published studies to define approximately 100 total rules that dictate the behavior of fiber cells, SSC, fibroblasts, neutrophils, and macrophages, as well as their interactions with the microenvironment, including microvasculature remodeling and cytokine diffusion and secretion (*Figure 7*). For a rule to be incorporated into the model, there had to be an established understanding within the literature supporting the behavior (i.e. multiple studies reporting similar findings or supported by other reputable publications). When available, we used experimental data to define the parameters associated with the model rules. There were 52 parameters that could not be related to known physiological measurement; therefore, these parameters were calibrated using parameter density estimation which will be described below in *Model calibration*. Following calibration of model parameters, separate model outputs were validated by comparison with experimental data, and various model perturbations were conducted and compared to literature results. This process allowed us to have confidence in the predictive capabilities of the model so that we could simulate and predict the sensitivity of muscle regeneration to changes in cytokines.

## Cellular-Potts modeling framework
Prior work to construct computational models to represent muscle recovery have used ordinary differential equation (*Stephenson and Kojouharov, 2018*) or agent-based modeling (ABM) software, such as Netlogo (*Martin et al., 2016*) or Repast (*Virgilio et al., 2021*). While these models have yielded great insights into skeletal muscle damage and recovery processes, they have limited capacity to represent the spatial diffusion of cytokines accurately and explicitly throughout the skeletal muscle. The Cellular-Potts model framework (*Swat et al., 2012*) (CPM, also known as the Glazier-Graner-Hogeweg model), proved an ideal choice because it allows for logic-based representation of cellular behavior and interactions characteristic of agent-based modeling (ABM) (see *Supplementary file 3* for CPM mathematical implementation, *Supplementary file 4* for CPM adhesion parameters).

## ABM design
The ABM spatially represents a 2D male murine skeletal muscle fascicle cross-section of approximately 50 muscle fibers (*Figure 1*). The ABM depicts the microenvironment of the cross-section as well as the spatial migration of cells and diffusion of various cytokines (*Supplementary file 5*). The ABM simulates the emergent phenomenon of muscle tissue from an acute injury over the course of 28 days. The spatial agents in the model include muscle fibers, necrotic muscle tissues, ECM, capillaries, lymphatic vessels, quiescent and activated fibroblasts, myofibroblasts, quiescent and activated SSCs, myoblasts, myocytes, immature myotubes, neutrophils, monocytes, resident macrophages, pro-inflammatory macrophages (M1), and anti-inflammatory macrophages (M2). In addition, the ABM includes seven diffusing factors, such as HGF, MCP-1, MMP-9, TGF-β, TNF-α, VEGF-A, and IL-10. A review of the literature led us to determine that these factors and cytokine isoforms were most critical for representing the behaviors of each cell during the regeneration cascade (*Waldemer-Streyer et al., 2022*; *Rucavado et al., 2002*).

The muscle cross-section geometry was created by importing a histology image stained with laminin α2 into a custom MATLAB script that masked the histology image to distinguish between the fibers and ECM. The mask was imported into an initialization CC3D script that defined the muscle fibers, ECM, and microvasculature to specific cell types and generated a PIF file that was imported into the ABM as the starting cross-section. The injury is simulated by stochastically selecting a region within the cross-section to replace the fiber elements with necrotic elements, where the percentage of CSA damage is an input parameter. When a threshold of fiber elements within a muscle fiber becomes damaged, the entire muscle fiber turns necrotic and requires clearance. If the damage is below the threshold, only the region of necrosis must be removed and the SSCs can fuse to the remaining fiber. During model initialization, the injury criteria can be altered to simulate various degrees of myotoxin injury by changing the percent of necrotic tissue following injury.

Each Monte Carlo step (mcs) represents a 15 min timestep, and the model simulations were run until 28 days post injury. The cell velocity is limited by how many times the Cellular-Potts algorithm is run, so we set 45 Cellular-Potts evaluations per mcs to ensure stability in migratory agent behavior. The number of Cellular-Potts evaluations per mcs and the lambda chemotaxis parameters were tuned in a simplified simulation of individual cells and their respective chemotactic gradients so we could

**Table 1.** Neutrophil agent rules.

| Neutrophil agent behavior | Sources |
|---|---|
| Recruitment signal: necrosis | *Madaro and Bouché, 2014* |
| Neutrophils are brought to site of injury via capillaries | *Wang et al., 2020* |
| Phagocytose necrosis | *Butterfield et al., 2006* |
| Secretes MMP-9, MCP-1, TNF-α during phagocytosis | *Martin et al., 2016*; *Madaro and Bouché, 2014*; *Wang, 2018*; *Soehnlein et al., 2008* |
| Undergoes apoptosis after phagocytosis or 12.5 hr | *Fox et al., 2010* |
| Migrates toward areas of high HGF | *Molnarfi et al., 2015* |
| Migration speed ~7.5 µm/min | *Zhao et al., 2020*; *Heit et al., 2008* |

obtain cell speeds that were consistent with speeds derived from literature sources (Table 7). At each mcs, the agent behaviors are governed by rules that were derived from experimental data found in the literature. The behaviors of each agent are based on environmental conditions, such as nearby cells and cytokine gradients, as well as probability-based rules. As an example, a capillary located near a damaged fiber has a probability of becoming non-perfused and then senses the amount of VEGF-A and MMP-9 at its location to decide if the levels are adequate to induce angiogenesis (Table 6). Model outputs include CSA recovery (sum of total healthy fiber elements normalized by the initial CSA), capillary and collagen density, cell counts, relative cytokine abundance, and spatial coordinates of cells and cytokines.

**Table 2.** Macrophage agent rules.

| Macrophage agent behavior | Sources |
|---|---|
| Initial count: 1 resident macrophage per 5 myofibers | *Oishi and Manabe, 2018* |
| Recruitment signal: MCP-1 | *Vogel et al., 2014*; *Chazaud et al., 2003* |
| Monocytes are brought to the site of injury via microvessels | *Kratofil et al., 2017* |
| Resident macrophages secrete MMP-9, MCP-1, and TNF-α and chemotax along MCP-1 and HGF | *Elkington et al., 2009*; *Chen and Nuñez, 2010*; *Lacy and Stow, 2011*; *Vogel et al., 2014*; *Molnarfi et al., 2015*; *Furrer and Handschin, 2017* |
| Monocytes chemotax along MCP-1, VEGF-A, and TGF-β | *Chazaud et al., 2003*; *Owen and Mohamadzadeh, 2013*; *Reibman et al., 1991*; *Martin et al., 2017* |
| Monocyte migration speed ~4 µm/min | *van den Bos et al., 2020* |
| M1 macrophages secrete VEGF-A, MMP-9, and TNF-α and chemotax along MCP-1 | *Corliss et al., 2016*; *Newby, 2008*; *Lu et al., 2018*; *Cui et al., 2018* |
| Monocytes, resident, and M1 macrophages phagocytose apoptotic neutrophils and necrosis | *Greenlee-Wacker, 2016*; *Watanabe et al., 2019*; *Uribe-Querol and Rosales, 2020* |
| Monocytes and macrophages secrete MMP-9, HGF, TGF-β, and IL-10 during phagocytosis | *Martin et al., 2016*; *Yoon et al., 2016*; *D'Angelo et al., 2013*; *Popov et al., 2010*; *Arnold et al., 2007*; *Chung et al., 2007* |
| Monocyte transitions into M1 occurs when TNF-α threshold is met or based on literature means and standard deviation properties | *Arnold et al., 2007*; *Mosser and Edwards, 2008* |
| M1 transition into M2 is mediated by the amount of IL-10 and the amount the M1 has phagocytosed | *Martin et al., 2016*; *Arnold et al., 2007*; *Saini et al., 2016*; *Das et al., 2015* |
| M2 macrophages secrete TGF-β and IL-10 and chemotax along MCP-1 | *Martin et al., 2016*; *Vogel et al., 2014*; *Arabpour et al., 2021*; *da Silva et al., 2015* |
| Macrophages can proliferate following the transition to the anti-inflammatory (M2) state | *Arnold et al., 2007* |
| Macrophage migration speed ~0.62 µm/min | *van den Bos et al., 2020* |
| Macrophages apoptose in a Poisson distribution | *Moncayo, 2007* |

**Table 3.** SSC agent rules.

| SSC agent behavior | Sources |
|---|---|
| Initial count: 1 SSC per 4 fibers | *Virgilio et al., 2018*; *Reimann et al., 2000* |
| Recruitment signal: HGF + MMP-9 - TGF-β | *Virgilio et al., 2018*; *Kawamura et al., 2004*; *Wang et al., 2009*; *Allen and Boxhorn, 1989*; *González et al., 2017* |
| Activation signal: HGF | *Virgilio et al., 2018*; *González et al., 2017*; *Allen et al., 1995*; *Miller et al., 2000*; *Tatsumi et al., 1998* |
| Activated SSCs secrete MCP-1 and VEGF-A | *Chazaud et al., 2003* |
| Activated SSCs migrate toward areas of high MMP-9 | *Wang et al., 2009*; *Chen and Li, 2009* |
| Myoblasts migrate toward high TNF-α | *Torrente et al., 2003* |
| Division signal: TNF-α + VEGF-A - TGF-β | *Virgilio et al., 2018*; *Allen and Boxhorn, 1989*; *Bakkar et al., 2008*; *Saclier et al., 2013* |
| Differentiation signal: 3*IL-10 - HGF - TNF-α - TGF-β | *Virgilio et al., 2018*; *Saini et al., 2016*; *Perandini et al., 2018*; *Gal-Levi et al., 1998*; *Ten Broek et al., 2010* |
| Activated SSCs differentiate into myoblasts, myoblasts into myocytes, and myocytes into myotubes/myofibers | *Cooper et al., 1999*; *Flamini et al., 2018*; *Bentzinger et al., 2012* |
| Differentiated myocytes fuse at damaged fiber edge or fuse together to form new, immature myotubes | *Yin et al., 2013*; *Wang et al., 2014*; *Nguyen et al., 2019*; *Ruiz-Gómez et al., 2002* |
| 50% cell divisions are symmetric, 50% asymmetric | *Virgilio et al., 2018*; *Kuang et al., 2007*; *Yennek et al., 2014* |
| Division probability decreases with each cell division; first division 85%; second 65%; third 20% | *Virgilio et al., 2018*; *Siegel et al., 2011* |
| VEGF-A and macrophages nearby can block apoptosis | *Chazaud et al., 2003*; *Arsic et al., 2004*; *Sonnet et al., 2006* |
| TGF-β triggers apoptosis | *Cencetti et al., 2013* |
| Time to divide: 10 hr | *Virgilio et al., 2018*; *Siegel et al., 2011*; *Rocheteau et al., 2012* |
| Migration speed ~0.94 μm/min | *Otto et al., 2011* |
| Return activated SSCs to quiescence without sustained HGF | *González et al., 2017* |

## Overview of agent behaviors

Simulated behaviors (*Figure 1B*) of the neutrophils and macrophages include cytokine-dependent recruitment, chemotaxis, phagocytosis of damaged fibers (neutrophils, monocytes, and M1 macrophages), phagocytosis of apoptotic neutrophils (monocytes and M1 macrophages), secretion and uptake of cytokines, and apoptosis. The SSC and fibroblast agent behaviors also include cytokine-dependent recruitment, chemotaxis, secretion and uptake of cytokines, and apoptosis, in addition to quiescence, activation, division, and differentiation. The biological intricacy of some cell types, such as SSCs which have a more complex cell cycle and are regulated by dynamic interplay of intrinsic factors and an array of microenvironmental stimuli, led to the necessity for adding more rules that govern their behaviors (*Yin et al., 2013*).The neutrophils have 18 parameters for 7 agent rules (*Table 1*), macrophages have 31 parameters for 15 agent rules (*Table 2*), SSCs have 33 parameters dictating the 17 agent rules (*Table 3*), fibroblasts have 27 parameters for 11 agent rules (*Table 4*), fibers have 18 parameters for 4 agent rules (*Table 5*), and microvessels have 22 parameters for 6 agent rules (*Table 6*). At each mcs, cytokines are secreted by agents if certain conditions were met. For cell recruitment, the levels of recruiting cytokines for each agent are checked, and if the concentration is high enough to signal cell recruitment, a new agent is added to the field at the location of the highest concentration. The agents also undergo chemotaxis by sensing the surrounding cytokine gradients and move toward higher concentrations of cytokines, binding and removing that cytokine as they move along it to simulate physical binding of the cytokine to the receptor. Agents that are in a quiescent state require a certain threshold level of cytokines to become activated and cannot chemotax, secrete, divide, or differentiate until this threshold is reached. Our model assumes each unique cell type secretes the same concentration of cytokines per timestep for all relevant cytokines

**Table 4.** Fibroblast agent rules.

| Fibroblast agent behavior | Sources |
|---|---|
| Initial count: 1 fibroblast per every 2 fibers | *Virgilio et al., 2018*; *Murphy et al., 2011* |
| Activation signal: TGF-β | *Gibb et al., 2020* |
| Fibroblasts move to low collagen ECM | *Virgilio et al., 2018*; *Dickinson et al., 1994* |
| Fibroblasts secrete TNF-α, TGF-β, MMP-9, VEGF-A. Collagen is secreted at low-density ECM | *Virgilio et al., 2018*; *Zou et al., 2008*; *Sanderson et al., 1986*; *Yokoyama et al., 1999*; *Skutek et al., 2001*; *Lindner et al., 2012*; *Newman et al., 2011* |
| Fibroblast division signaled by SSC division | *Virgilio et al., 2018*; *Murphy et al., 2011* |
| Division probability decreases with each cell division; first division 100%; second 25%; third 6% | *Alberts et al., 2002* |
| Fibroblast differentiation into myofibroblasts with extended TGF-β exposure | *Virgilio et al., 2018*; *Desmoulière et al., 1993*; *Wipff et al., 2007* |
| Myofibroblasts secrete double the amount of collagen and secretion is not dependent on collagen density | *Virgilio et al., 2018*; *Petrov et al., 2002* |
| Fibroblasts apoptose with sustained exposure to TNF-α | *Virgilio et al., 2018*; *Lemos et al., 2015* |
| Fibroblast migration speed ~0.73 µm/min | *Cornwell et al., 2004* |
| Sufficient TGF-β can block fibroblast apoptosis | *Virgilio et al., 2018*; *Lemos et al., 2015* |

to drive model agent decisions. Each computational timestep represents 15 min of real-world time. We assume that this is of sufficient resolution to accurately reproduce immune cell agent behaviors during regeneration.

## Neutrophil agents

Neutrophils are recruited through capillaries to sites of necrotic tissue (*Table 1*). Neutrophils move to areas of necrotic tissue with high concentrations of HGF by chemotaxing along the HGF gradient to reach areas of necrosis (*Madaro and Bouché, 2014*; *Wang et al., 2020*). Neutrophils phagocytose necrotic tissue and facilitate remodeling into ECM with low collagen density. During phagocytosis, neutrophils secrete MMP-9, MCP-1, and TNF-α (*Butterfield et al., 2006*; *Madaro and Bouché, 2014*; *Wang, 2018*; *Soehnlein et al., 2008*). Individual neutrophil agents apoptose after phagocytosing two necrotic cells (based on calibration) or 12.5 hr after their recruitment (*Fox et al., 2010*).

## Macrophage agents

Resident macrophages are distributed randomly throughout the tissue at a ratio of 1 macrophage per 5 myofibers at model initialization and secrete MCP-1 (*Oishi and Manabe, 2018*; *Table 2*). Resident macrophages chemotax along MCP-1 and HGF chemical gradients and secrete MMP-9, TNF-α, and MCP-1 during simulation (*Elkington et al., 2009*; *Chen and Nuñez, 2010*; *Lacy and Stow, 2011*; *Vogel et al., 2014*; *Molnarfi et al., 2015*; *Furrer and Handschin, 2017*). After tissue injury, monocytes are recruited through healthy capillary microvasculature and chemotax along MCP-1, VEGF-A, TGF-β (*Kratofil et al., 2017*; *Chazaud et al., 2003*; *Owen and Mohamadzadeh, 2013*; *Reibman et al., 1991*). Monocytes infiltrate into the tissue if the MCP-1 concentration is above a specified threshold at a capillary site. Resident macrophages, monocytes, and the M1 macrophages differentiated from

**Table 5.** Fiber agent rules.

| Fiber agent behavior | Sources |
|---|---|
| Damaged muscle fibers secrete HGF and TGF-β | *Miller et al., 2000*; *Kim and Lee, 2017* |
| Healthy fibers secrete VEGF-A | *Huey, 2018* |
| Fibers that are fully necrotic are fusion incompetent, but damaged fibers are fusion competent | *Snijders et al., 2015* |
| Immature myotubes gain functional capacity as they fully mature over time | *Nguyen et al., 2019*; *Abmayr and Pavlath, 2012*; *Isesele and Mazurak, 2021* |

**Table 6.** Microvasculature rules.

| Microvessel agent behavior | Sources |
| --- | --- |
| Initial count: ~4 capillaries per fiber, 1 lymphatic vessel per fascicle | *Wickler, 1981*; *Gehlert et al., 2010* |
| Capillaries near necrosis will become damaged and unable to perfuse | *Jacobsen et al., 2021* |
| With sufficient VEGF-A damaged capillaries will undergo angiogenesis | *Frey et al., 2012* |
| MMP-9 is elevated during capillary growth | *Haas et al., 2000*; *Qutub et al., 2009* |
| Increasing capillary-to-myofiber ratio during muscle regeneration from new sprouting capillaries at areas with enough MMP-9 and VEGF-A | *Jacobsen et al., 2021*; *Hardy et al., 2016*; *Haas et al., 2000* |
| Cells and cytokines near lymphatic vessel will be drained via the vessel and removed from microenvironment | *Hampton and Chtanova, 2019* |

monocytes may phagocytose areas of necrotic tissue and apoptotic neutrophil agents (*Greenlee-Wacker, 2016*; *Watanabe et al., 2019*; *Uribe-Querol and Rosales, 2020*). During phagocytosis, these agents secrete MMP-9, HGF, TGF-β, and IL-10 (*Yoon et al., 2016*; *D'Angelo et al., 2013*; *Popov et al., 2010*; *Arnold et al., 2007*; *Chung et al., 2007*).

Monocytes transition to M1 polarized macrophages when the monocyte agent experiences a large enough TNF-α concentration or if enough time has passed that a predefined transition time threshold is met. Each monocyte agent at creation has a defined transition time sampled from a Gaussian distribution with mean and SD set to reproduce literature-defined populations of M1 macrophages over time (*Arnold et al., 2007*; *Mosser and Edwards, 2008*).

M1 macrophages may transition to M2 macrophages if the M1 macrophage agent experiences an IL-10 concentration that exceeds a threshold value or if the M1 macrophage has phagocytosed enough to meet a calibrated threshold value (as discussed in *Model calibration*) (*Arnold et al., 2007*; *Saini et al., 2016*; *Das et al., 2015*). Following the transition to the anti-inflammatory phenotype, the M2 macrophages can proliferate, secrete TGF-β and IL-10, and chemotax along an MCP-1 gradient (*Vogel et al., 2014*; *Arnold et al., 2007*; *Arabpour et al., 2021*).

## SSC agents

The model is initialized with 1 quiescent SSC per every 4 fibers and upon injury (*Reimann et al., 2000*). Additional SSCs are recruited based on the amount of HGF, MMP-9, and TGF-β (*Kawamura et al., 2004*; *Wang et al., 2009*; *Allen and Boxhorn, 1989*; *González et al., 2017*; *Table 3*). For SSC activation there has to be enough HGF at the location of the quiescent SSC to induce activation (*González et al., 2017*; *Allen et al., 1995*; *Miller et al., 2000*; *Tatsumi et al., 1998*). The SSCs also chemotax up the MMP-9 gradient, removing some of the MMP-9 as they move along it. Activated SSCs can also undergo symmetric or asymmetric division and differentiation given that the required cytokine signaling is met locally. Activated SSCs differentiated into myoblasts and myoblasts differentiate into myocytes (*Cooper et al., 1999*; *Flamini et al., 2018*; *Bentzinger et al., 2012*). Myocytes can fuse to other myocytes to form new myotubes or fuse to fibers as long as the fiber is not fusion incompetent (i.e. fully necrotic) (*Yin et al., 2013*; *Wang et al., 2014*; *Nguyen et al., 2019*; *Ruiz-Gómez et al., 2002*). Maturation of myotubes is required for fusion of additional myocytes to the new fiber (*Nguyen et al., 2019*; *Abmayr and Pavlath, 2012*; *Isesele and Mazurak, 2021*). If the damage signal is not sustained, activated SSCs return to quiescence. If there is enough TGF-β to induce apoptosis and not enough VEGF-A or macrophages nearby to block it, the SSC undergoes cell death and leaves the simulation (*Chazaud et al., 2003*; *Arsic et al., 2004*; *Sonnet et al., 2006*; *Cencetti et al., 2013*).

## Fibroblast agents

For model initialization, fibroblasts are randomly placed within the ECM at a population size that is proportional to the number of fibers (*Murphy et al., 2011*; *Table 4*). Fibroblasts are activated based on the concentration of TGF-β around the fibroblast (*Beanes et al., 2003*; *Chellini et al., 2019*). Fibroblasts include an additional expression in their effective energy function that directs their migration toward areas of low-density collagen ECM (*Dickinson et al., 1994*). Specifically, fibroblasts can form spring-like links to drag them toward areas of low-density ECM which are implemented with the

relation $\lambda_{ij}\left(l_{ij} - L_{ij}\right)^2$, where $\lambda_{ij}$ denotes a Hookean spring constant of a link between cells $i$ and $j$, $l$ represents the current distance between the centers of mass between the two cells (in our case, fibroblast and low collagen ECM), and $L$ is the target length of the spring-like link. In addition to the cytokines secreted by fibroblasts (*Table 4*), collagen is secreted at low-density collagen ECM (*Zou et al., 2008*; *Sanderson et al., 1986*; *Yokoyama et al., 1999*; *Skutek et al., 2001*; *Lindner et al., 2012*; *Newman et al., 2011*). Fibroblasts divide when they are near dividing SSCs and can differentiate into myofibroblasts with extended exposure to TGF-β (*Murphy et al., 2011*; *Desmoulière et al., 1993*; *Wipff et al., 2007*). The myofibroblasts can secrete more collagen regardless of the ECM density (*Petrov et al., 2002*). Fibroblasts can undergo apoptosis if there are adequate levels of TNF-α at the site of the cell but it can be blocked if there is sufficient TGF-β (*Lemos et al., 2015*).

## ECM agents

ECM elements surround the fiber elements and are assigned a collagen density parameter which varies based on the amount of necrotic tissue removed and the extent of fibroblast/myofibroblast collagen secretion. When necrotic elements are removed, the phagocytosing inflammatory cells secrete MMP-9s which degrade some of the collagen within that section of the ECM, thereby causing that element to have a lower collagen density (*Madaro and Bouché, 2014*). The collagen density of the ECM alters the diffusivity of the secreted factors, and fiber placement is dependent on the collagen density (discussed below). The fibroblasts help rebuild the ECM by secreting collagen on low collagen density ECM elements (*Zou et al., 2008*). Myofibroblasts can secrete collagen on any ECM element and if prolonged results in high-density collagen elements, representing a fibrotic state.

## Fiber and necrotic agents

Upon model initialization, a portion of the muscle fiber agents are converted to necrotic fibers based on the user prescribed injury. Fibers that reach a damaged threshold became fully necrotic whereas those surrounding the area of necrosis were damaged but not fully apoptotic cells. Healthy fiber elements secrete VEGF-A, and necrotic elements secrete HGF and TGF-β (*Miller et al., 2000*; *Kim and Lee, 2017*; *Huey, 2018*; *Table 5*). Phagocytosing agents chemotax along those gradients to clear the necrosis, but before a new fiber can be deposited, the collagen has to be restored so that there is a scaffold to hold the fiber in place (*Oishi and Manabe, 2018*). Fully necrotic fibers are fusion incompetent and require myocyte-to-myocyte fusion to form a new myofiber and require maturation before additional myocyte fusion (*Nguyen et al., 2019*; *Abmayr and Pavlath, 2012*; *Isesele and Mazurak, 2021*). Damaged fibers are regenerated by myocytes fusion to the healthy fiber edge (*Snijders et al., 2015*).

## Capillary and lymphatic agents

The muscle fascicle environment includes approximately 4 capillaries per fiber and 1 lymphatic vessel (*Wickler, 1981*; *Gehlert et al., 2010*; *Table 6*). The model defines perfused capillaries as capillary agents that can transport neutrophils and monocytes into the system proportional to the concentration of recruiting cytokines (*Wang et al., 2020*; *Kratofil et al., 2017*). The neutrophils and monocytes are added to the simulation at the lattice sites above capillaries (within the cell layer; *Figure 1B*) and chemotax along their respective gradients. The recruitment of the neutrophils and monocytes are distributed among the healthy capillaries with a higher affinity for capillaries at locations with higher concentrations of HGF and MCP-1, respectively. Under physiologically reasonable chemotactic gradient conditions, the recruited immune cells dispersed efficiently, with no aggregation. Capillaries that are neighboring areas of necrosis become non-perfused and therefore are unable to transport cells into the microenvironment until regenerated (*Jacobsen et al., 2021*). Angiogenesis can occur as long as there is enough VEGF-A present at the non-perfused capillary (*Frey et al., 2012*). Similar to published studies, there is an increase in the capillary-to-myofiber ratio during muscle regeneration, which is due to the formation of new capillary sprouts modulated in part by local MMP-9 and VEGF-A levels (*Jacobsen et al., 2021*; *Hardy et al., 2016*; *Haas et al., 2000*).

The lymphatic vessel uptakes cytokines at lattice locations corresponding to the lymphatic vessel and will remove cells located in lattice sites neighboring those corresponding to the lymphatic vessel (*Hampton and Chtanova, 2019*). In addition, we have included a rule in our ABM to encourage cells to migrate toward the lymphatic vessel utilizing CompuCell3D External Potential Plugin

**Table 7.** Model parameters of spatial mechanisms.

| Parameter | Value | Source/justification |
|---|---|---|
| *Volume parameters* | | |
| Target volume neutrophil | 12 | Chosen for an average cell diameter of 12 µm (*Tigner et al., 2021*) |
| Target volume SSC | 10 | Chosen for an average cell diameter of 10 µm (*Garcia et al., 2018*) |
| Target volume macrophage | 21 | Chosen for an average cell diameter of 21 µm (*Krombach et al., 1997*) |
| Target volume monocyte | 8.5 | Chosen for an average cell diameter of 8.5 µm (*Downey et al., 1990*) |
| Target volume fibroblast | 15 | Chosen for an average cell diameter of 15 µm (*Freitas, 1999*) |
| Volume multiplier $\lambda_{volume}$ | 50 | Volume constraint to maintain target (*Swat et al., 2012*) |
| *Diffusion coefficients* | | |
| HGF | 66.38 µm$^2$/s | |
| MMP-9 | 63.40 µm$^2$/s | |
| MCP-1 | 189.27 µm$^2$/s | |
| VEGF-A | 112.10 µm$^2$/s | |
| TGF-β | 90.33 µm$^2$/s | |
| TNF-α | 138.95 µm$^2$/s | Estimated diffusivity within the ECM accounting for baseline GAGs and collagen (*Filion and Popel, 2005*) |
| IL-10 | 135.17 µm$^2$/s | |
| *Chemotaxis parameters $\lambda_c$* | | |
| Neutrophils | 750 | Chosen for a cell velocity between 1 and 20 µm/min (*Zhao et al., 2020*) |
| Macrophage | 9.3 | Chosen for a cell velocity around 0.62 µm/min (*van den Bos et al., 2020*) |
| Monocyte | 75 | Chosen for a cell velocity around 4 µm/min (*van den Bos et al., 2020*) |
| SSC | 11.3 | Chosen for a cell velocity around 0.94 µm/min (*Otto et al., 2011*) |
| Fibroblast | 23 | Chosen for a cell velocity around 0.73 µm/min (*Westman et al., 2021*) |

(*ExternalPotential Plugin, 2024*). The influence of this rule is inversely proportional to the distance of the cells to the lymphatic vessel.

## Binding, diffusivity, and collagen density

For many of the agent behaviors described above, there are associated binding events that play key roles in regulation of the cytokine fields. Any cytokine-dependent behavior is coupled with removal of a portion of that cytokine once the behavior is initiated. For example, upon SSC activation the amount of HGF required to activate is taken up by the SSC and removed from the cytokine field to simulate the ligand binding and endocytosis resulting from SSC activation. Similar binding events were modeled for SSC and fibroblast division and differentiation, macrophage transitions, cell apoptosis, and chemotaxis along a cytokine gradient.

Due to limited data availability quantifying the diffusion constants of the modeled cytokines in the context of the tissue microenvironment (which includes diffusion-altering elements including collagen and glycosaminoglycans [GAGs]), we applied a diffusivity estimation technique (*Filion and Popel, 2005*). To do so, previously developed methods (*Equation 1*) were applied to account for the

combined effects of collagen and GAGs (*Table 7*; *Filion and Popel, 2005*). The expression includes the radius of the cytokine ($r_s$), the radius of the fiber ($r_f$), the volume fraction ($\phi$), $D$ and $D_\infty$ are the diffusivities of the cytokines in the polymer solution and in free solution, respectively. This estimation technique allowed for consistent conditions for cytokine diffusion calculations and fluctuations based on changes in collagen density within the model.

$$D = D_\infty \left( -\phi^{\frac{1}{2}} \frac{r_s}{r_f} \right)_{collagen} \times exp \left( -\phi^{\frac{1}{2}} \frac{r_s}{r_f} \right)_{GAG} \tag{1}$$

Throughout the model simulation, the diffusivity is recalculated with the updated collagen volume fraction, as the collagen density changes throughout the microenvironment. This allows the changes in collagen density within the ECM to be reflected in the diffusion rate of each of the cytokines in the model.

## Model calibration

Known parameters were fixed to literature values, and uncertain parameters were calibrated by comparing simulation outcomes to published experimental data. Calibration data included published findings from injury models that have synchronous regeneration after tissue necrosis (i.e. cardiotoxin, notexin, and barium chloride) (*Hardy et al., 2016*). The metrics that were used to calibrate the model included time-varying CSA (*Ochoa et al., 2007*), SSC counts (*Murphy et al., 2011*), and fibroblast counts (*Murphy et al., 2011*). These metrics were used for calibration because of their key roles in the regeneration of muscle and the complex interplay between these outputs. Cell count data were normalized by the number of cells on the day of the experimental peak to allow for comparison between experiments and simulations. For CSA, the experimental and model outcomes were normalized using fold-change from pre-injury to compare model-simulated with experimental CSA, as percent change from baseline is commonly used experimentally (*Pratt et al., 2015*; *You et al., 2023*). Model cell counts were normalized by the number of cells at the peak timepoint in the experimental data. SSC and fibroblast counts were normalized to day 5. Neutrophil counts were normalized to day 1. Total macrophage, M1, and M2 counts were normalized to day 3. The capillaries were normalized to fiber area, as done in the experimental data.

Initial ranges for the 52 unknown parameters were determined by literature review or by running the model to test possible upper and lower thresholds for parameters (*Supplementary file 1*). To narrow the parameter ranges beyond those initial ranges, we used a recently published calibration protocol, CaliPro, which utilizes parameter density estimation to refine parameter space and calibrate to temporal biological datasets (*Joslyn et al., 2021*). CaliPro was selected as the calibration method because it is model-agnostic which allows it to handle the complexities of stochastic models such as ABMs, selects viable parameter ranges in the setting of a very high-dimensional parameter space, and circumvents the need for a cost function, a challenge when there are many objectives, as in our case. Briefly, Latin hypercube sampling (LHS) was used to generate 600 samples which were run in triplicate. These runs were then evaluated against a set of pass criteria, and the density functions of the passing runs and failing runs were calculated (*Supplementary file 6*). Parameter ranges were narrowed by alternative density subtraction, where the new ranges were determined by the smallest and largest parameter values where the density of passing is higher than the density of failing. The sensitivity of the model outputs to the parameters was examined using LHS in combination with PRCC (*Marino et al., 2008*). LHS/PRCC methods have been used for various differential equation models and ABMs (*Segovia-Juarez et al., 2004*). PRCC was computed using MATLAB to determine the correlation between ABM parameters (i.e. cytokine threshold for activation) and the ABM output (i.e. fibroblast cell count). Correlations with a p-value less than 0.05 were assumed to be statistically significant. This helped refine initial parameter bounds as well as make model adjustments based on the parameter dynamics elucidated from PRCC. This process of sampling parameter ranges, evaluating the model, and narrowing parameter ranges was repeated in an iterative fashion while updating pass criteria until a parameter set was identified that consistently met the strictest criteria (*Figure 2—figure supplement 1*). The final passing criteria were set to be within 1 SD of the experimental data for CSA recovery and 2.5 SD for SSC and fibroblast count. These criteria were selected so that the model followed experimental trends and accounted for both model stochasticity and experimental variability

**Table 8.** Model perturbation input conditions and corresponding published experimental results.

| Perturbation | Specific model conditions | Published outcomes |
|---|---|---|
| IL-10 knockout | Adjust diffusion and decay parameters so IL-10 is removed from the system | Attenuates shift to M2, disrupted SSC differentiation, slowed regeneration (*Deng et al., 2012*) |
| Neutrophil depletion | Lower neutrophil recruitment proportion | Abundant necrotic tissue 7 days post injury (*Teixeira et al., 2003*) |
| Macrophage depletion | Lower macrophage recruitment proportion | Decreased HGF, increased TGF-β and TNF-α, impaired regeneration (*Liu et al., 2017*) |
| MCP-1 knockout | Adjust diffusion and decay parameters so MCP-1 is removed from the system | Increased necrosis at day 7, lower CSA at day 21, impaired phagocytosis (*Lu et al., 2011*) |
| Directed M2 polarization (anti-inflammatory nanoparticles) | Require less phagocytosis and IL-10 for transition | Improved muscle histology and inflammatory resolution (*Raimondo and Mooney, 2018*) |
| TNF-α knockout | Adjust diffusion and decay parameters so TNF-α is removed from the system | Impaired recovery at days 5 and 12, increased inflammation (*Chen et al., 2005*) |
| Hindered angiogenesis | Increase VEGF-A and MMP-9 threshold required for angiogenesis | Delayed regeneration with toxin injury, and persistent immune cell infiltration with freeze injury (*Hardy et al., 2019*) |
| VEGF-A injection | Add VEGF-A at specified concentration (100 for low and 1000 relative concentration for high), radius (300 pixels), and timepoint (5 days post injury) | Lower injury area at day 20 post injury with injection 5 days after damage (*Arsic et al., 2004*) |

in datasets that had narrower SDs for certain timepoints. Early iterations had a wide parameter range to avoid missing portions of the realistic parameter space. At first, narrowing the parameter space increased passing simulations, but upon reaching the ideal parameter space, further narrowing eliminated viable parameters, resulting in fewer passing runs. Following eight iterations of narrowing the parameter space with CaliPro, we reached a set of parameters that had fewer passing runs than the previous iteration. We then returned to the runs from the prior iteration and set the bounds such that all three runs from the parameter set fell within the final passing criteria. The final parameter set was run 100 times to verify that the variation from the stochastic nature of the rules did not cause output that was inconsistent with experimental trends.

## Model validation

We compared model outputs M1, M2, and total macrophage counts (*Hardy et al., 2016*; *Wang et al., 2018*), neutrophil counts (*Nguyen et al., 2011*), and capillary counts (*Ochoa et al., 2007*) that were kept separate from the calibration criteria with published experimental data to verify that these outputs followed trends from the experimental data without requiring extra model tuning. In addition, we also altered various model input conditions (cell input conditions, cytokine dynamics, and microvessel dynamics) to simulate an array of model perturbations (*Table 8*) which allowed comparison of a set of model outputs with separate published experiments (*Supplementary file 7*). For example, we simulated an IL-10 KO condition by eliminating IL-10 secretion and adjusting the diffusion and decay parameters so that the concentration of IL-10 throughout the simulation was reduced, decreasing the behaviors driven by the cytokine as a result of the KO condition. One hundred replicates of each model perturbation were performed, and perturbation outputs were compared with control simulation outputs via a two-sample t-test with a significance level of 0.05. We were then able to compare how the model outputs aligned with published experimental findings to determine if the model could capture the altered regeneration dynamics.

## Sensitivity analysis

A sensitivity analysis was performed using LHS-PRCC to examine the impact of cytokine-related parameters on model outputs of interest. Diffusion coefficients and decay rates for the seven cytokines (HGF, TGF-β, MMP-9, TNF-α, VEGF-A, IL-10, MCP-1) were sampled across a range from 0.1 to 10 times the calibrated value while holding the other parameters constant. Three hundred samples were generated, and these parameter sets were simulated in triplicate. PRCCs were calculated with $\alpha$=0.05 and a Bonferroni correction for the number of tests every 10 ticks/hr for CSA and cell counts

**Table 9.** Summary of cytokine sensitivity analysis.

Significance was determined with $\alpha=0.05$, and a Bonferroni correction for the number of tests. + and - represent statistically significant positive and negative correlations, respectively.

| | CSA | SSC | Fibroblasts | Non-perfused capillaries | Myoblasts | Myocytes | Neutrophils | M1 | M2 |
|---|---|---|---|---|---|---|---|---|---|
| Day | 16.7 | 6.3 | 10.5 | 8.4 | 6.3 | 8.4 | 8.4 | 4.2 | 6.3 |
| HGF decay | - | - | - | + | - | - | + | | + |
| TGF-β decay | + | + | + | - | + | | | | - |
| MMP-9 decay | + | + | + | - | + | + | | | |
| TNF-α decay | | | + | | | | | | - |
| VEGF-A decay | | | | + | | | | | |
| MCP-1 decay | | | | | | | | + | + |
| MCP-1 diffusion | | + | | - | | | | + | |

for SSCs, fibroblasts, non-perfused capillaries, myoblasts, myocytes, neutrophils, M1 macrophages, and M2 macrophages.

## In silico experiments

To gain insight into the recovery response with altered angiogenesis, we simulated different levels of VEGF-A injections to test how increases in VEGF-A impacted regeneration outcomes. In addition, we simulated conditions of hindered angiogenesis in which damaged capillaries were unable to reperfuse following injury ($n=100$ for each simulation condition). Simulations were also conducted to examine correlations between cytokines and their impact on various cell behaviors and regeneration outcomes. Next, a sensitivity analysis was performed to understand how alterations in cytokines influence key metrics of regeneration. LHS-PRCC was used to quantify the impact of cytokine-related parameters (i.e. diffusion rates and decay coefficients) on outputs of interest (CSA, SSC, fibroblasts, non-perfused capillaries, myoblasts, myocytes, neutrophils, M1, and M2). A single timepoint for each output is summarized in *Table 9*, and these were chosen at the timepoint when PRCC values were peaking, with complete results available in *Figure 6—figure supplement 1*.

This sensitivity analysis was then used to guide in silico experiments based on which cytokine parameters promoted favorable regeneration outcomes (i.e. improved recovery, fewer non-perfused capillaries, increased SSCs). Following individual cytokine parameter alterations, we combined the cytokine alterations based on beneficial outcomes from the initial in silico experiments to determine if the benefits would be cumulative.

## Acknowledgements

The authors acknowledge NIH Grant #R21AR080415, Wu-Tsai Foundation Agility Project Funding, and NSF GRFP Grant #1842490 for financially supporting this research and to James Glazier and TJ Sego for providing technical support with the CC3D ABM platform.

## Additional information

### Funding

| Funder | Grant reference number | Author |
|---|---|---|
| National Institutes of Health | R21AR080415 | Silvia S Blemker |
| National Science Foundation | 1842490 | Megan Haase |
| Wu-Tsai Foundation | Agility Project Funding | Silvia S Blemker |

| Funder | Grant reference number | Author |
| --- | --- | --- |

The funders had no role in study design, data collection and interpretation, or the decision to submit the work for publication.

## Author contributions

Megan Haase, Conceptualization, Data curation, Software, Formal analysis, Funding acquisition, Validation, Investigation, Visualization, Methodology, Writing - original draft, Project administration, Writing – review and editing; Tien Comlekoglu, Data curation, Software, Formal analysis, Investigation, Methodology, Writing - original draft, Writing – review and editing; Alexa Petrucciani, Software, Formal analysis, Investigation, Methodology, Writing - original draft, Writing – review and editing; Shayn M Peirce, Resources, Funding acquisition, Investigation, Methodology, Writing – review and editing; Silvia S Blemker, Conceptualization, Resources, Supervision, Funding acquisition, Investigation, Methodology, Writing – review and editing

## Author ORCIDs

Megan Haase http://orcid.org/0000-0002-5221-4495
Shayn M Peirce http://orcid.org/0000-0001-5857-5606
Silvia S Blemker https://orcid.org/0000-0002-2019-1153

Reviewer #1 (Public Review): https://doi.org/10.7554/eLife.91924.3.sa1
Reviewer #2 (Public Review): https://doi.org/10.7554/eLife.91924.3.sa2
Author response https://doi.org/10.7554/eLife.91924.3.sa3

# Additional files

## Supplementary files

• Supplementary file 1. Unknown model parameters calibrated using Latin hypercube sampling (LHS) to recapitulate published literature.

• Supplementary file 2. Cytokine perturbations based on partial rank correlation coefficient (PRCC).

• Supplementary file 3. Cellular-Potts model (CPM) mathematical implementation.

• Supplementary file 4. Cellular-Potts model (CPM) agent adhesion parameters.

• Supplementary file 5. Cellular-Potts model (CPM) initialization model parameters.

• Supplementary file 6. Criteria utilized for CaliPro model calibration.

• Supplementary file 7. Experimental data description for model comparison.

• MDAR checklist

## Data availability

The ABM source code is publicly available at the following sites: SimTK; Zendo; GitHub (copy archived at *mh2uk, 2024*).

The following datasets were generated:

| Author(s) | Year | Dataset title | Dataset URL | Database and Identifier |
| --- | --- | --- | --- | --- |
| Haase M, Petrucciani A, Comlekoglu T, Peirce S, Blemker S | 2024 | Agent-Based Model of Muscle Regeneration with Microvascular Remodeling | https://zenodo.org/doi/10.5281/zenodo.10403013 | Zenodo, 10.5281/zenodo.10403013 |
| Haase M, Petrucciani A, Comlekoglu T, Peirce S, Blemker S | 2024 | Agent-Based Model of Muscle Regeneration with Microvascular Remodeling | https://simtk.org/docman/?group_id=2635 | SimTK, 2635 |

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
