## [Editor Report · eLife assessment]

This is so-far the most comprehensive, spatially resolved in 2D, dynamical, multicellular model of murine muscle regeneration after injury. The work is an attempt to combine many contributors to muscle regeneration into one coherent calibrated framework. The presented analysis is **solid** and the model has the potential to be a very **valuable** tool in the areas of tissue morphogenesis, regenerative therapies, quantitative modeling and simulation.

---

## [Referee Report · Reviewer #1 (Public Review)]

Summary:

This work extends previous agent-based models of murine muscle regeneration by the authors (especially Westman et al., 2021) and by others (especially Khuu et al, 2023) by incorporating additional agent rules (altogether now based on over 100 published studies), threshold parameters and interactions with fields of cytokines and growth factors as well as capillaries (dynamically changing through damage and angiogenesis) and lymphatic vessels. The estimation of 52 unknown parameters against three time courses of tissue-scale observables (muscle cross-sectional area recovery, satellite stem cell count and fibroblast cell count) employs the CaliPro algorithm (Joslyn et al., 2021) and sensitivity analysis. The model is validated against additional time courses of tissue-scale observables and qualitative perturbation data, which match almost all conditions. This model is here used to predict (also non-monotonic) responses of (combinations of) cytokine perturbations but it moreover represents a valuable resource for further analysis of emergent behavior across multiple spatial scales in a physiologically relevant system.

Strengths:

This work (almost didactically) demonstrates how to develop, calibrate, validate and analyze a comprehensive, spatially resolved, dynamical, multicellular model. Testable model predictions of (also non-monotonic) emergent behaviors are derived and discussed. The computational model is based on a widely-used simulation platform and shared openly such that it can be further analyzed and refined by the community. The single-used parameter set is a good starting point for future work that can, as outlined in the discussion section of the paper, analyze model results from the full distribution of matching parameter values and for a spectrum of realistic tissue configurations.

---

## [Referee Report · Reviewer #2 (Public Review)]

Summary:

In the paper, the authors use a cellular Potts model to investigate muscle regeneration. The model is an attempt to combine many contributors to muscle regeneration into one coherent framework. I believe the resulting model has the potential to be very useful in investigating the complex interplay of multiple actors contributing to muscle regeneration.

Strengths:

The manuscript identified relevant model parameters from a long list of biological studies. This collation of a large amount of literature into one framework has the potential to be very useful to other authors. The mathematical methods used for parameterization and validation are transparent.

Comments on revised version:

The authors have satisfactorily addressed my previous comments.

---

## [Author Response]

The following is the authors’ response to the original reviews.

**Public Reviews:**

**Reviewer #1 (Public Review):**
Strengths:This work (almost didactically) demonstrates how to develop, calibrate, validate and analyze a comprehensive, spatially resolved, dynamical, multicellular model. Testable model predictions of (also non-monotonic) emergent behaviors are derived and discussed. The computational model is based on a widely-used simulation platform and shared openly such that it can be further analyzed and refined by the community.Weaknesses:While the parameter estimation approach is sophisticated, this work does not address issues of structural and practical non-identifiability (Wieland et al., 2021,DOI:10.1016/j.coisb.2021.03.005) of parameter values, given just tissue-scale summary statistics, and does not address how model predictions might change if alternative parameter combinations were used. Here, the calibrated model represents one point estimate (column "Value" in Suppl. Table 1) but there is specific uncertainty of each individual parameter value and such uncertainties need to be propagated (which is computationally expensive) to the model predictions for treatment scenarios.

We thank the reviewer for the excellent suggestions and observations. The CaliPro parameterization technique applied puts an emphasis on finding a robust parameter space instead of a global optimum. To address structural non-identifiability, we utilized partial rank correlation coefficient with each iteration of the calibration process to ensure that the sensitivity of each parameter was relevant to model outputs. We also found that there were ranges of parameter values that would achieve passing criteria but when testing the ranges in replicate resulted in inconsistent outcomes. This led us to further narrow the parameters into a single parameter set that still had stochastic variability but did not have such large variability between replicate runs that it would be unreliable. Additional discussion on this point has been added to lines 623-628. We acknowledge that there are likely other parameter sets or model rules that would produce similar outcomes but the main purpose of the model was to utilize it to better understand the system and make new predictions, which our calibration scheme allowed us to accomplish.

Regarding practical non-identifiability, we acknowledge that there are some behaviors that are not captured in the model because those behaviors were not specifically captured in the calibration data. To ensure that the behaviors necessary to answer the aims of our paper were included, we used multiple different datasets and calibrated with multiple different output metrics. We believe we have identified the appropriate parameters to recapitulate the dominating mechanisms underlying muscle regeneration.We have added additional discussion on practical non-identifiability to lines 621-623.

Suggested treatments (e.g. lines 484-486) are modeled as parameter changes of the endogenous cytokines (corresponding to genetic mutations!) whereas the administration of modified cytokines with changed parameter values would require a duplication of model components and interactions in the model such that cells interact with the superposition of endogenous and administered cytokine fields. Specifically, as the authors also aim at 'injections of exogenously delivered cytokines' (lines 578, 579) and propose altering decay rates or diffusion coefficients (Fig. 7), there needs to be a duplication of variables in the model to account for the coexistence of cytokine subtypes. One set of equations would have unaltered (endogenous) and another one have altered (exogenous or drugged) parameter values. Cells would interact with both of them.

Our perturbations did not include delivery of exogenously delivered cytokines and instead were focused on microenvironmental changes in cytokine diffusion and decay rates or specific cytokine concentration levels. For example, the purpose of the VEGF delivery perturbation was to test how an increase in VEGF concentrations would alter regeneration outcome metrics with the assumption that the delivered VEGF would act in the same manner as the endogenous VEGF. We have clarified the purpose of the simulations on line 410. We agree that exploring if model predictions would be altered if endogenous and exogenous were represented separately; however, we did not explore this type of scenario.

This work shows interesting emergent behavior from nonlinear cytokine interactions but the analysis does not provide insights into the underlying causes, e.g. which of the feedback loops dominates early versus late during a time course.

Indeed, analyzing the model to fully understand the time-varying interactions between the multiple feedback loops is a challenge in and of itself, and we appreciate the opportunity to elaborate on our approach to addressing this challenge. First: the crosstalk/feedback between cytokines and the temporal nature was analyzed in the heatmap (Fig. 6) and lines 474-482. Second: the sensitivity of cytokine parameters to specific outputs was included in Table 9 and full-time course sensitivity is included in Supplemental Figure 2. Further correlation analysis was also included to demonstrate how cytokine concentrations influenced specific output metrics at various timepoints (Supplemental Fig. 3). We agree that further elaboration of these findings is required; therefore, we added lines 504-509 to discuss the specific mechanisms at play with the combined cytokine interactions. We also added more discussion (lines 637-638) regarding future work that could develop more analysis methods to further investigate the complex behaviors in the model.

**Reviewer #2 (Public Review):**
Strengths:The manuscript identified relevant model parameters from a long list of biological studies. This collation of a large amount of literature into one framework has the potential to be very useful to other authors. The mathematical methods used for parameterization and validation are transparent.Weaknesses:>I have a few concerns which I believe need to be addressed fully.My main concerns are the following:(1) The model is compared to experimental data in multiple results figures. However, the actual experiments used in these figures are not described. To me as a reviewer, that makes it impossible to judge whether appropriate data was chosen, or whether the model is a suitable descriptor of the chosen experiments. Enough detail needs to be provided so that these judgements can be made.

Thank you for raising this point. We created a new table (Supplemental table 6) that describes the techniques used for each experimental measurement.

(2) Do I understand it correctly that all simulations are done using the same initial simulation geometry? Would it be possible to test the sensitivity of the paper results to this geometry? Perhaps another histological image could be chosen as the initial condition, or alternative initial conditions could be generated in silico? If changing initial conditions is an unreasonably large request, could the authors discuss this issue in the manuscript?

We appreciate your insightful question regarding the initial simulation geometry in our model. The initial configuration of the fibers/ECM/microvascular structures was kept consistent but the location of the necrosis was randomly placed for each simulation. Future work will include an in-depth analysis of altered histology configuration on model predictions which has been added to lines 618-621. We did a preliminary example analysis by inputting a different initial simulation geometry, which predicted similar regeneration outcomes. We have added Supplemental Figure 5 that provides the results of that example analysis.

(3) Cytokine knockdowns are simulated by 'adjusting the diffusion and decay parameters' (line 372). Is that the correct simulation of a knockdown? How are these knockdowns achieved experimentally? Wouldn't the correct implementation of a knockdown be that the production or secretion of the cytokine is reduced? I am not sure whether it's possible to design an experimental perturbation which affects both parameters.

We appreciate that this important question has been posed. Yes, in order to simulate the knockout conditions, the cytokine secretion was reduced/eliminated. The diffusion and decay parameters were also adjusted to ensure that the concentration within the system was reduced. Lines 391-394 were added to clarify this assumption.

(4) The premise of the model is to identify optimal treatment strategies for muscle injury (as per the first sentence of the abstract). I am a bit surprised that the implemented experimental perturbations don't seem to address this aim. In Figure 7 of the manuscript, cytokine alterations are explored which affect muscle recovery after injury. This is great, but I don't believe the chosen alterations can be done in experimental or clinical settings. Are there drugs that affect cytokine diffusion? If not, wouldn't it be better to select perturbations that are clinically or experimentally feasible for this analysis? A strength of the model is its versatility, so it seems counterintuitive to me to not use that versatility in a way that has practical relevance. - I may well misunderstand this though, maybe the investigated parameters are indeed possible drug targets.

Thank you for your thoughtful feedback. The first sentence (lines 32-34) of the abstract was revised to focus on beneficial microenvironmental conditions to best reflect the purpose of the model. The clinical relevance of the cytokine modifications is included in the discussion (lines 547-558) with additional information added to lines 524-526. For example, two methods to alter diffusion experimentally are: antibodies that bind directly to the cytokine to prevent it from binding to its receptor on the cell surface and plasmins that induce the release of bound cytokines.

(5) A similar comment applies to Figure 5 and 6: Should I think of these results as experimentally testable predictions? Are any of the results surprising or new, for example in the sense that one would not have expected other cytokines to be affected as described in Figure 6?

We appreciate the opportunity to clarify the basis for these perturbations. The perturbations included in Figure 5 were designed to mimic the conditions of a publishedexperiment that delivered VEGF in vivo (Arsic et al. 2004, DOI:10.1016/J.YMTHE.2004.08.007). The perturbation input conditions and experimental results are included in Table 8 and Supplemental Table 6 has been added to include experimental data and method description of the perturbation. The results of this analysis provide both validation and new predictions, because some the outputs were measured in the experiments while others were not measured. The additional output metrics and timepoints that were not collected in the experiment allow for a deeper understanding of the dynamics and mechanisms leading to the changes in muscle recovery (lines 437-454). These model outputs can provide the basis for future experiments; for example, they highlight which time points would be more important to measure and even provide predicted effect sizes that could be the basis for a power analysis (lines 639-640).

Regarding Figure 6, the published experimental outcomes of cytokine KOs are included in Table 8. The model allowed comparison of different cytokine concentrations at various timepoints when other cytokines were removed from the system due to the KO condition. The experimental results did not provide data on the impact on other cytokine concentrations but by using the model we were able to predict temporally based feedback between cytokines (lines 474-482). These cytokine values could be collected experimentally but would be time consuming and expensive. The results of these perturbations revealed the complex nature of the relationship between cytokines and how removal of one cytokine from the system has a cascading temporal impact. Lines 533-534 have been added to incorporate this into the discussion.

(6) In figure 4, there were differences between the experiments and the model in two of the rows. Are these differences discussed anywhere in the manuscript?

We appreciate your keen observation and the opportunity to address these differences. The model did not match experimental results for CSA output in the TNF KO and antiinflammatory nanoparticle perturbation or TGF levels with the macrophage depletion. While it did align with the other experimental metrics from those studies, it is likely that there are other mechanisms at play in the experimental conditions that were not captured by simulating the downstream effects of the experimental perturbations. We have added discussion of the differences to lines 445-454.

(7) The variation between experimental results is much higher than the variation of results in the model. For example, in Figure 3 the error bars around experimental results are an order of magnitude larger than the simulated confidence interval. Do the authors have any insights into why the model is less variable than the experimental data? Does this have to do with the chosen initial condition, i.e. do you think that the experimental variability is due to variation in the geometries of the measured samples?

Thank you for your insightful observations and questions. The lower model variability is attributed to the larger sample size of model simulations compared to experimental subjects. By running 100 simulations it narrows in the confidence interval (average 2.4 and max 3.3) compared to the experiments that typically had a sample size of less than 15. If the number of simulations had been reduced to 15 the stochasticity within the model results in a larger confidence interval (average 7.1 and max 10). There are also several possible confounding variables in the experimental protocols (i.e. variations in injury, different animal subjects for each timepoint, etc.) that are kept constant in the model simulation. We have added discussion of this point to the manuscript (lines 517519). Future work with the model will examine how variations in conditions, such as initial muscle geometry, injury, etc, alter regeneration outcomes and overall variability. This discussion has been incorporated into lines 640-643.

(8) Is figure 2B described anywhere in the text? I could not find its description.

Thank you for pointing that out. We have added a reference for Fig. 2B on line 190.

**Recommendations for the authors:**

**Reviewer #1 (Recommendations For The Authors):**
(1) The model code seems to be available from https://simtk.org/projects/muscle_regen but that website requests member status ("This is a private project. You must be a member to view its contents.") and applying for membership could violate eLife's blind review process. So, this reviewer liked to but couldn't run the model her/himself. To eLife: Can the authors upload their model to a neutral server that reviewers and editors can access anonymously?

The code has been made publicly available on the following sites:

SimTK: https://simtk.org/docman/?group_id=2635

Zendo: https://zenodo.org/records/10403014

GitHub: https://github.com/mh2uk/ABM-of-Muscle-Regeneration-with-MicrovascularRemodeling

Line 121 has been updated with the new link and the additional resources were added to lines 654-657.

(2) The muscle regeneration field typically studies 2D cross-sections and the present model can be well compared to these other 2D models but cells as stochastic and localized sources of diffusible cytokines may yield different cytokine fields in 3D vs. 2D. I would expect more broadened and smoothened cytokine fields (from sources in neighboring cross-sections) than what the 2D model predicts based on sources just within the focus cross-section. Such relations of 2D to 3D should be discussed.

We thank the reviewer for the excellent suggestions and observations. It has been reported in other Compucell3D models (Sego et al. 2017, DOI:10.1088/17585090/aa6ed4) that the convergence of diffusion solutions between 2D and 3D model configurations had similar outcomes, with the 3D simulations presenting excessive computational cost without contributing any noticeable additional accuracy. Similarly, other cell-based ABMs that incorporate diffusion mechanisms (Marino et al. 2018, DOI:10.3390/computation6040058) have found that 2D and 3D versions of the model both predict the same mechanisms and that the 2D resolution was sufficient for determining outcomes. Lines 615-618 were added to elaborate on this topic.

(3) Since the model (and title) focuses on "nonlinear" cytokine interactions, what would change if cytokine decay would not be linear (as modeled here) but saturated (with nonlinear Michaelis-Menten kinetics as ligand binding and endocytosis mechanisms would call for)?

Thank you for raising an intriguing point. The model includes a combination of cytokine decay as well as ligand binding and endocytosis mechanisms that can be saturated. For a cytokine-dependent model behavior to occur the cytokines necessary to induce that action had to reach a minimum threshold. Once that threshold was reached, that amount of the cytokine would be removed at that location to simulate ligand-receptor binding and endocytosis. These ligand binding and endocytosis mechanisms behave in a saturated way, removing a set amount when above a certain threshold or a defined ratio when under the threshold. Lines 313-315 was revised to clarify this point. There were certain concentrations of cytokines where we saw a plateau in outputs likely as a result of reaching a saturation threshold (Supplemental Fig. 3). In future work, more robust mathematical simulation of binding kinetics of cytokines (e.g., using ODEs) could be included.

(4) Limitations of the model should be discussed together with an outlook for model refinement. For example, fiber alignment and ECM ultrastructure may require anisotropic diffusion. Many of the rate equations could be considered with saturation parameters etc. There are so many model assumptions. Please discuss which would be the most urgent model refinements and, to achieve these, which would be the most informative next experiments to perform.

We appreciate your thoughtful consideration of the model's limitations and the need for a comprehensive discussion on model refinements and potential future experiments. The future direction section was expanded to discuss additional possible model refinements (lines 635-643) and additional possible experiments for model validation (lines 630-634).

(5) It is not clear how the single spatial arrangement that is used affects the model predictions. E.g. now the damaged area surrounds the lymphatic vessel but what if the opposite corner was damaged and the lymphatic vessel is deep inside the healthy area?

Thank you for highlighting the importance of considering different spatial arrangements in the model and its potential impact on predictions. We previously tested model perturbations that included specifying the injury surrounding the lymphatic vessel versus on the side opposite the vessel. Since this paper focuses more on cytokine dynamics, we plan to include this perturbation, along with other injury alterations, in a follow-on paper. We added more context about this in the future efforts section lines 640-643.

(6) It seems that not only parameter values but also the initial values of most of the model components are unknown. The parameter estimation strategy does not seem to include the initial (spatial) distributions of collagen and cytokines and other model components. Please discuss how other (reasonable) initial values or spatial arrangements will affect model predictions.

We appreciate your thoughtful consideration of unknown initial values/spatial arrangements and their potential influence on predictions. Initial cytokine levels prior to injury had a low relative concentration compared to levels post injury and were assumed to be negligible. Initial spatial distribution of cytokines was not defined as initial spatial inputs (except in knockout simulations) but are secreted from cells (with baseline resident cell counts defined from the literature). The distribution of cytokines is an emergent behavior that results from the cell behaviors within the model. The collagen distribution is altered in response to clearance of necrosis by the immune cells (decreased collagen with necrosis removal) and subsequent secretion of collagen by fibroblasts. The secretion of collagen from fibroblast was included in the parameter estimation sweep (Supplemental Table 1).

We are working on further exploring the model sensitivity to altered spatial arrangements and have added this to the future directions section (lines 618-621), as well as provided Supplemental Figure 5 to demonstrate that model outcomes are similar with altered initial spatial arrangements.

(7) Many details of the CC3D implementation are missing: overall lattice size, interaction neighborhood order, and "temperature" of the Metropolis algorithm. Are the typical adhesion energy terms used in the CPM Hamiltonian and if so, then how are these parameter values estimated?

Thank you for bringing attention to the missing details regarding the CC3D implementation in our manuscript. We have included supplemental information providing greater detail for CPM implementation (Lines 808-854). We also added two additional supplemental tables for describing the requested CC3D implementation details (Supplemental Table 4) and adhesion energy terms (Supplemental Table 5).

(8) Extending the model analysis of combinations of altered cytokine properties, which temporal schedules of administration would be of interest, and how could the timing of multiple interventions improve outcomes? Such a discussion or even analysis would further underscore the usefulness of the model.

In response to your valuable suggestion, lines 558-562 were added to discuss the potential of using the model as a tool to perturb different cytokine combinations at varying timepoints throughout regeneration. In addition, this is also included in future work in lines 636-637.

(9) The CPM is only weakly motivated, just one sentence on lines 142-145 which mentions diffusion in a misleading way as the CPM just provides cells with a shape and mechanical interactions. The diffusion part is a feature of the hybrid CompuCell3D framework, not the CPM.

Thank you for bringing up this distinction. We removed the statement regarding diffusion and updated lines 143-146 to focus on CPM representation of cellular behavior and interactions. We also added a reference to supplemental text that includes additional details on CPM.

(10) On lines 258-261 it does not become clear how the described springs can direct fibroblasts towards areas of low-density collagen ECM. Are the lambdas dependent on collagen density?

Thank you for highlighting this area for clarification. The fibroblasts form links with low collagen density ECM and then are pulled towards those areas based on a constant lambda value. The links between the fibroblast and the ECM will only be made if the collagen is below a certain threshold. We added additional clarification to lines 260-264.

(11) On line 281, what does the last part in "Fibers...were regenerating but not fully apoptotic cells" mean? Maybe rephrase this.

The last of part of that line indicates that there were some fibers surrounding the main injury site that were damaged but still had healthy portions, indicating that they were impacted by the injury and are regenerating but did not become fully apoptotic like the fiber cells at the main site of injury. We rephrased this line to indicate that the nearby fibers were damaged but not fully apoptotic.

(12) Lines 290-293 describe interactions of cells and fields with localized structures (capillaries and lymphatic vessel). Please explain in more detail how "capillary agents...transport neutrophiles and monocytes" in the CPM model formalism. Are new cells added following rules? How is spatial crowding of the lattice around capillaries affecting these rules? Moreover, how can "lymphatic vessel...drain the nearby cytokines and cells"? How is this implemented in the CPM and how is "nearby" calculated? We appreciate your detailed inquiry into the interactions of cells and fields with localized structures. The neutrophils and monocytes are added to the simulation at the lattice sites above capillaries (within the cell layer Fig. 2B) and undergo chemotaxis up their respective gradients. The recruitment of the neutrophils and monocytes are randomly distributed among the healthy capillaries that do not have an immune cell at the capillary location (a modeling artifact that is a byproduct of only having one cell per lattice site). This approach helped to prevent an abundance of crowding at certain capillaries. Because immune cells in the simulation are sufficiently small, chemotactic gradients are sufficiently large, and the simulation space is sufficiently large, we do not see aggregation of recruited immune cells in the CPM.

The lymphatic vessel uptakes cytokines at lattice locations corresponding to the lymphatic vessel and will remove cells located in lattice sites neighboring the lymphatic vessel. In addition, we have included a rule in our ABM to encourage cells to migrate towards the lymphatic vessel utilizing CompuCell3D External Potential Plugin. The influence of this rule is inversely proportional to the distance of the cells to the lymphatic vessel.

We have updated lines 294-298 and 305-309 to include the above explanation.

(13) Tables 1-4 define migration speeds as agent rules but in the typical CPM, migration speed emerges from random displacements biased by chemotaxis and other effects (like the slope of the cytokine field). How was the speed implemented as a rule while it is typically observable in the model?

We appreciate your inquiry regarding the implementation of migration speeds. To determine the lambda parameters (Table 7) for each cell type, we tested each in a simplified control simulation with a concentration gradient for the cell to move towards. We tuned the lambda parameters within this simulation until the model outputted cell velocity aligned with the literature reported cell velocity for each cell type (Tables 1-4). We have incorporated clarification on this to lines 177-180.

(14) Line 312 shows the first equation with number (5), either add eqn. (1-4) or renumber.

We have revised the equation number.

(15) Typos: Line 456, "expect M1 cell" should read "except M1 cell".Line 452, "thresholds above that diminish fibroblast response (Supplemental Fig 3)." remains unclear, please rephrase.Line 473, "at 28." should read "at 28 days.".Line 474, is "additive" correct? Was the sum of the individual effects calculated and did that match?Line 534, "complexity our model" should read "complexity in our model".

We have corrected the typos and clarified line 452 (updated line 594) to indicate that the TNF-α concentration threshold results in diminished fibroblast response. We updated terminology line 474 (updated line 512) to indicate that there was a synergistic effect with the combined perturbation.

(16) Table 7 defines cell target volumes with the same value as their diameter. This enforces a strange cell shape. Should there be brackets to square the value of the cell diameter, e.g. Value=(12µm)^2 ?

The target volume parameter values were selected to reflect the relative differences in average cell diameter as reported in the literature; however, there are no parameters that directly enforce a diameter for the cells in the CPM formalism separate from the volume. We have observed that these relative cell sizes allow the ABM to effectively reproduce cell behaviors described in the literature. Single cells that are too large in the ABM would be unable to migrate far enough per time step to carry out cell behaviors, and cells that are too small in the CPM would be unstable in the simulation environment and not persist in the simulation when they should. We removed the units for the cell shape values in Table 7 since the target volume is a relative parameter and does not directly represent µm.

(17) Table 7 gives estimated diffusion constants but they appear to be too high. Please compare them to measured values in the literature, especially for MCP-1, TNF-alpha and IL-10, or relate these to their molecular mass and compare to other molecules like FGF8 (Yu et al. 2009, DOI:10.1038/nature08391).

We utilized a previously published estimation method (Filion et al. 2004,DOI:10.1152/ajpheart.00205.2004) for estimating cytokine diffusivity within the ECM. This method incorporates the molecular masses and accounts for the combined effects of the collagen fibers and glycosaminoglycans. The paper acknowledged that the estimated value is faster than experimentally determined values, but that this was a result of the less-dense matrix composition which is more reflective of the tissue environment we are simulating in contrast to other reported measurements which were done in different environments. Using this estimation method also allowed us to more consistently define diffusion constants versus using values from the literature (which were often not recorded) that had varied experimental conditions and techniques (such as being in zebrafish embryo Yu et al. 2009, DOI:10.1038/nature08391 as opposed to muscle tissue). This also allowed for recalculation of the diffusivity throughout the simulation as the collagen density changed within the model. Lines 318-326 were updated to help clarify the estimation method.

(18) Many DOIs in the bibliography (Refs. 7,17,20,31,40,47...153) are wrong and do not resolve because the appended directory names are not allowed in the DOI, just with a journal's URL after resolution.

Thank you for bringing this to our attention. The incorrect DOIs have been corrected.

**Reviewer #2 (Recommendations For The Authors):**
Minor comments:(9) On line 174, the authors say "We used the CC3D feature Flip2DimRatio to control the number of times the Cellular-Potts algorithm runs per mcs." What does this mean? Isn't one monte carlo timestep one iteration of the Cellular Potts model? How does this relate to physical timescales?

We appreciate your attention to detail and thoughtful question regarding the statement about the use of the CC3D feature Flip2DimRatio. Lines 175-177 were revised to simplify the meaning of Flip2DimRatio. That parameter alters the number of times the Cellular-Potts algorithm is run, which is the limiting factor for cell movement. The physical timescale is kept to a 15-minute timestep but a high Flip2DimRatio allows more flexibility and stability to allow the cells to move faster in one timestep.

(10) Has the costum matlab script to process histology images into initial conditions been made available?

The Matlab script along with CC3D code for histology initialization with documentation has been made available with the source code on the following sites:

SimTK: https://simtk.org/docman/?group_id=2635

Zendo: https://zenodo.org/records/10403014

GitHub: https://github.com/mh2uk/ABM-of-Muscle-Regeneration-with-MicrovascularRemodeling

(11) Equation 5 is provided without a reference or derivation. Where does it come from and what does it mean?

Thank you for highlighting the diffusion equation and seeking clarification on its origin and significance. Lines 318-326 were revised to clarify where the equation comes from. This is a previously published estimation method that we applied to calculate the diffusivity of the cytokines considering both collagen and glycosaminoglycans.

(12) Line 326: "For CSA, experimental fold-change from pre-injury was compared with fold-change in model-simulated CSA". Does this step rely on the assumption that the fold change will not depend on the CSA? If so, is this something that is experimentally known, or otherwise, can it be confirmed by simulations?

We appreciate the opportunity to clarify our rationale. The fold change was used as a method to normalize the model and experiment so that they could be compared on the same scale. Yes, this step relies on the assumption that fold change does not depend on pre-injury CSA. Experimentally it is difficult to determine the impact of initial fiber morphology on altered regeneration time course. This fold-change allows us to compare percent recovery which is a common metric utilized to assess muscle regeneration outcomes experimentally. Line 340-343 was revised to clarify.

(13) Line 355: "The final passing criteria were set to be within 1 SD for CSA recovery and 2.5 SD for SSC and fibroblast count" Does this refer to the experimental or the simulated SD?

The model had to fit within those experimental SD. Lines 371-372 was edited to specify that we are referring the experimental SD.

(14) "Following 8 iterations of narrowing the parameter space with CaliPro, we reached a set that had fewer passing runs than the previous iteration". Wouldn't one expect fewer passing runs with any narrowing of the parameter space? Why was this chosen as the stopping criterion for further narrowing?

We appreciate your observation regarding the statement about narrowing the parameter space with CaliPro. We started with a wide parameter space, expecting that certain parameters would give outputs that fall outside of the comparable data. So, when the parameter space was narrowed to enrich parts that give passing output, initially the number of passing simulations increased.

Once we have narrowed the set of possible parameters into an ideal parameter space, further narrowing will cut out viable parameters resulting in fewer passing runs. Therefore, we stopped narrowing once any fewer simulations passed the criteria that they had previously passed with the wider parameter set. Lines 375-379 have been updated to clarify this point.

(15) Line 516: 'Our model could test and optimize combinations of cytokines, guiding future experiments and treatments." It is my understanding that this is communicated as a main strength of the model. Would it be possible to demonstrate that the sentence is true by using the model to make actual predictions for experiments or treatments?

This is demonstrated by the combined cytokine alterations in Figure 7 and discussed in lines 509-513. We have also added in a suggested experiment to test the model prediction in lines 691-695.

(16) Line 456, typo: I think 'expect' should be 'except'.

Thank you for pointing that out. The typo has been corrected.